# The miR-221/222 cluster regulates hematopoietic stem cell quiescence and multipotency by suppressing both Fos/AP-1/IEG pathway activation and stress-like differentiation to granulocytes

**Peter K. Jani** [1]*, **Georg Petkau**[1], **Yohei Kawano**[1], **Uwe Klemm**[2], **Gabriela Maria Guerra**[1], **Gitta Anne Heinz**[1], **Frederik Heinrich**[1], **Pawel Durek**[1], **Mir-Farzin Mashreghi**[1], **Fritz Melchers**[1,2]*

**1** Deutsches Rheuma Forschungszentrum (DRFZ), Berlin, Germany, **2** Max Planck Institute for Infection Biology, Berlin, Germany

* peter_karoly.jani@drfz.de (PKJ); fritz.melchers@unibas.ch (FM)

**Data Availability Statement:** All relevant data are within the paper and its supporting files. Raw data

## Abstract

Throughout life, hematopoietic stem cells (HSCs), residing in bone marrow (BM), continuously regenerate erythroid/megakaryocytic, myeloid, and lymphoid cell lineages. This steady-state hematopoiesis from HSC and multipotent progenitors (MPPs) in BM can be perturbed by stress. The molecular controls of how stress can impact hematopoietic output remain poorly understood. MicroRNAs (miRNAs) as posttranscriptional regulators of gene expression have been found to control various functions in hematopoiesis. We find that the miR-221/222 cluster, which is expressed in HSC and in MPPs differentiating from them, perturbs steady-state hematopoiesis in ways comparable to stress. We compare pool sizes and single-cell transcriptomes of HSC and MPPs in unperturbed or stress-perturbed, miR-221/222-proficient or miR-221/222-deficient states. MiR-221/222 deficiency in hematopoietic cells was induced in C57BL/6J mice by conditional vav-cre-mediated deletion of the floxed miR-221/222 gene cluster. Social stress as well as miR-221/222 deficiency, alone or in combination, reduced HSC pools 3-fold and increased MPPs 1.5-fold. It also enhanced granulopoisis in the spleen. Furthermore, combined stress and miR-221/222 deficiency increased the erythroid/myeloid/granulocytic precursor pools in BM. Differential expression analyses of single-cell RNAseq transcriptomes of unperturbed and stressed, proficient HSC and MPPs detected more than 80 genes, selectively up-regulated in stressed cells, among them immediate early genes (IEGs). The same differential single-cell transcriptome analyses of unperturbed, miR-221/222-proficient with deficient HSC and MPPs identified Fos, Jun, JunB, Klf6, Nr4a1, Ier2, Zfp36—all IEGs—as well as CD74 and Ly6a as potential miRNA targets. Three of them, Klf6, Nr4a1, and Zfp36, have previously been found to influence myelogranulopoiesis. Together with increased levels of *Jun, Fos* forms increased amounts of the heterodimeric activator protein-1 (AP-1), which is known to control the expression of the selectively up-regulated expression of the IEGs. The comparisons of single-cell mRNA-deep sequencing analyses of socially stressed with miR-221/222-deficient

are uploaded to public repositories. The single cell RNA-seq datasets are available through NCBI under the following GEO accession numbers GSE227322, GSE227505, GSE227520, respectively. Source data are provided with this paper. Normalized scRNA-seq data are available in S1 Data and S2 Data. Flow cytometry data are available through FlowRepository.org (ID: FR-FCM-Z6PS). The software used in this study is open source. Cellranger from 10xgenomics: (https://support.10xgenomics.com/single-cell-gene-expression/software/downloads/latest); Seurat packages 4.1.1: (https://cloud.r-project.org/web/packages/Seurat/index.html). Used scripts are available upon request. (https://CRAN.R-project.org/CRANlogo.png; https://cloud.r-project.org/web/packages/Seurat/index.html) CRAN-Package Seurat: (https://cloud.r-project.org/web/packages/Seurat/index.html), (cloud.r-project.org). R-scripts used to analyze the raw data are provided in S3 Data.

Funding: P.K.J was supported by the Postdoctoral Research Fellowship (HUN 1186808 HFST-P) in 2017-2018 got from the Alexander von Humboldt Foundation (https://www.humboldt-foundation.de). This work was supported by the Leibniz Collaborative Excellence Grant CHROQ-K121/2018 to FM (https://www.leibniz-gemeinschaft.de). The funders had no role in study design, data collection and analysis, decision to publish, or preparation of the manuscript.

Competing interests: The authors have declared that no competing interests exist.

Abbreviations: AP-1, activator protein-1; BM, bone marrow; CLP, common lymphoid progenitor; CMP, common myeloid progenitor; GMP, granulocyte-myelocyte progenitor; GSEA, gene set enrichment analysis; HSC, hematopoietic stem cell; IEG, immediate early gene; MEP, megakaryocyte-erythroid progenitor; miRNA, microRNA; MPP, multipotent progenitor; NK, natural killer; NTC, no template control; UMAP, Uniform Manifold Approximation and Projection for Dimension Reduction; UPR, unfolded protein response.

HSC identify 5 of the 7 Fos/AP-1-controlled IEGs, Ier2, Jun, Junb, Klf6, and Zfp36, as common activators of HSC from quiescence. Combined with stress, miR-221/222 deficiency enhanced the Fos/AP-1/IEG pathway, extended it to MPPs, and increased the number of granulocyte precursors in BM, inducing selective up-regulation of genes encoding heat shock proteins Hspa5 and Hspa8, tubulin-cytoskeleton-organizing proteins Tuba1b, Tubb 4b and 5, and chromatin remodeling proteins H3f3b, H2afx, H2afz, and Hmgb2. Up-regulated in HSC, MPP1, and/or MPP2, they appear as potential regulators of stress-induced, miR-221/222-dependent increased granulocyte differentiation. Finally, stress by serial transplantations of miR-221/222-deficient HSC selectively exhausted their lymphoid differentiation capacities, while retaining their ability to home to BM and to differentiate to granulocytes. Thus, miR-221/222 maintains HSC quiescence and multipotency by suppressing Fos/AP-1/IEG-mediated activation and by suppressing enhanced stress-like differentiation to granulocytes. Since miR-221/222 is also expressed in human HSC, controlled induction of miR-221/222 in HSC should improve BM transplantations.

## Introduction

Throughout life, hematopoietic stem cells (HSCs) reside in bone marrow (BM), from where they continue to generate, at steady, unperturbed state, all hematopoietic cell lineages [1–7]. When transplanted, HSCs reconstitute hematopoietic stem and progenitor cell compartments and all hematopoietic cell lineages in normal numbers in the host. At steady-state activation, HSCs enter cell cycle, and, when differentiating, give rise to multipotent progenitors (MPP1-MPP4), which generate common lymphoid (CLP) and to common myeloid progenitors (CMPs). HSCs are heterogeneous, with multilineage-potential, myeloid-erythroid-biased, myeloid-erythroid-restricted, and differentiation-inactive HSC [3,8,9]. How this heterogeneity is regulated is poorly understood.

Hematopoiesis can be perturbed by different forms of stress [1,4,10–24]. Before "ex vivo/in vitro" preparation of BM cells for FACS analyses, a 20-hour-long transportation of the mice from the breeding facility to the laboratory induces social stress [12]. This social stress is reversible by a 14- to 21-day rest period as it is recommended after housing mice in a new environment. "Ex vivo/in vitro" stress is probably only one of several forms of stress, which impact HSC, when they are transplanted repeatedly in serial transplantations to study their homing and reconstitution potential [1,4]. We compare pool sizes of HSC and of progenitor compartments (MPP1 and 2), as well as single-cell transcriptomic describing their gene expression programs, unperturbed at steady state with those perturbed by either social stress, "in vitro" stress, or transplantation.

MicroRNAs (miRNAs) as posttranscriptional regulators of gene expression have been shown to play a crucial role in hematopoiesis, modulating single or multiple genes in multiple cellular functions [25–30]. Specific ablation of Dicer in HSCs, thus complete deletion of mature miRNAs, leads to increased apoptosis in HSCs and affects their self-renewal capacity [30–32]. Several miRNAs have been identified, which influence the regulations of these different states of HSC [31,33] by the mTOR pathway [34,35].

The cluster of miR-221 and miR-222 was found to be expressed in early hematopoietic progenitors [35–37]. Except for a previous report that indicated a potential involvement of the miR-221/222 cluster in early erythropoiesis by regulating kit protein expression, the role of this microRNA cluster in early hematopoiesis remained unknown [38]. Furthermore, we have

previously described the influences of miR-221 on preBI-cell homing to BM [39]. MiR-221 overexpression confers the ability to these cells to home to, and become resident in special, subosteal areas in the BM after transplantation. This is accomplished by activation of the PI3K signaling network in response to BM niche factors like the chemokine CXCL12. This activation induces integrin VLA-4 to change to its high affinity binding conformation, which allows increased adhesion and residence in environments of BM expressing the cell adhesion molecule VCAM1 [39]. Since HSCs have the natural ability to home to and engraft in BM, we reasoned that miR-221/222 might influence HSC pools in BM by regulating their engraftment as in preB cells. As we show here, this hypothesis is not tenable for HSC.

Here, we show that all HSC and MPP express the miRNA cluster. We then generate mice, in which HSC and their subsequent lineages become targetable for deletion of the floxed miR-221/222 cluster, using Vav-promotor-induced expression of iCre [9,40]. Single-cell expression analyses show that all HSC and MPPs have their miR-221/222 cluster deleted.

Deletion of the miR-221/222 cluster results in reduced numbers of HSC during steady-state hematopoiesis, accompanied by increased numbers of MPPs (MPP1-MPP4) similar to the effect of social stress perturbation in miR-221/222-proficien mice. Importantly, the number of granulocytes in spleen is increased by the miR-221/222 deficiency.

MiR-221/222 deficiency combined with social stress further enhances granulopoiesis by increasing numbers of granulocyte progenitors in BM.

In order to understand the molecular mechanisms of stress perturbation and/or miR-221/222 deficiency, we generate a single-cell transcriptional landscape of hematopoietic progenitors in BM.

Single-cell transcriptomes of combined socially stressed, miR-221/222-deficient HSC and MPPs identified selectively up-regulated expression of genes encoding heat shock proteins, tubulin-cytoskeleton-organizing proteins, and chromatin remodeling proteins. Finally, combined stress induced by serial transplantations and miR-221/222 deficiency exhausts lymphoid differentiation, but not BM-homing and granulocyte differentiation capacities of HSC.

## Results

### Controlled perturbation of steady-state hematopoiesis by "in vivo" social stress reduces HSC pools and increases MPPs in BM

Social stress imposed for less than 1 day permitted us to measure and to compare the actions of social stress on pool sizes of miR-221/222-proficient BM cells. We found that within 24 hours, social stress reduced the HSC pool in BM 2-fold and increased MPP2 two-and-a-half-fold and MPP4 2.2-fold (**Fig 1A**; for nomenclature, see Wilson and colleagues and Cabezas-Wallscheid and colleagues [5,7]). We also analyzed if the effect of transportation-induced social stress on stem cell populations of BM is reverted to steady state by time. For this, we analyzed the numbers of cells 3 days, 7 days, and 21 days after transportation. We found that, with significant fluctuations, the number of HSC, MPP1, MPP2, MPP3, and MPP4 becomes indistinguishable from the unperturbed pool of cells 3 weeks after social stress induction (**Fig 1A**). We did not detect significant differences at any time points after social stress perturbation in numbers of common lymphoid (CLP), megakaryocyte-erythroid (MEP), common myeloid (CMP), and granulocyte-myelocyte (GMP) progenitors in BM of miR-221/222-proficient mice (**S1A Fig**).

### Perturbation by "in vivo" social stress induces increased transcription of selected genes, among them IEG in HSC, MPP1, and MPP2 cells

Next, we focused our transcriptome analyses on the earliest progenitors HSC, MPP1, and MPP2 to monitor early changes in gene expression, imposed by the change from the steady

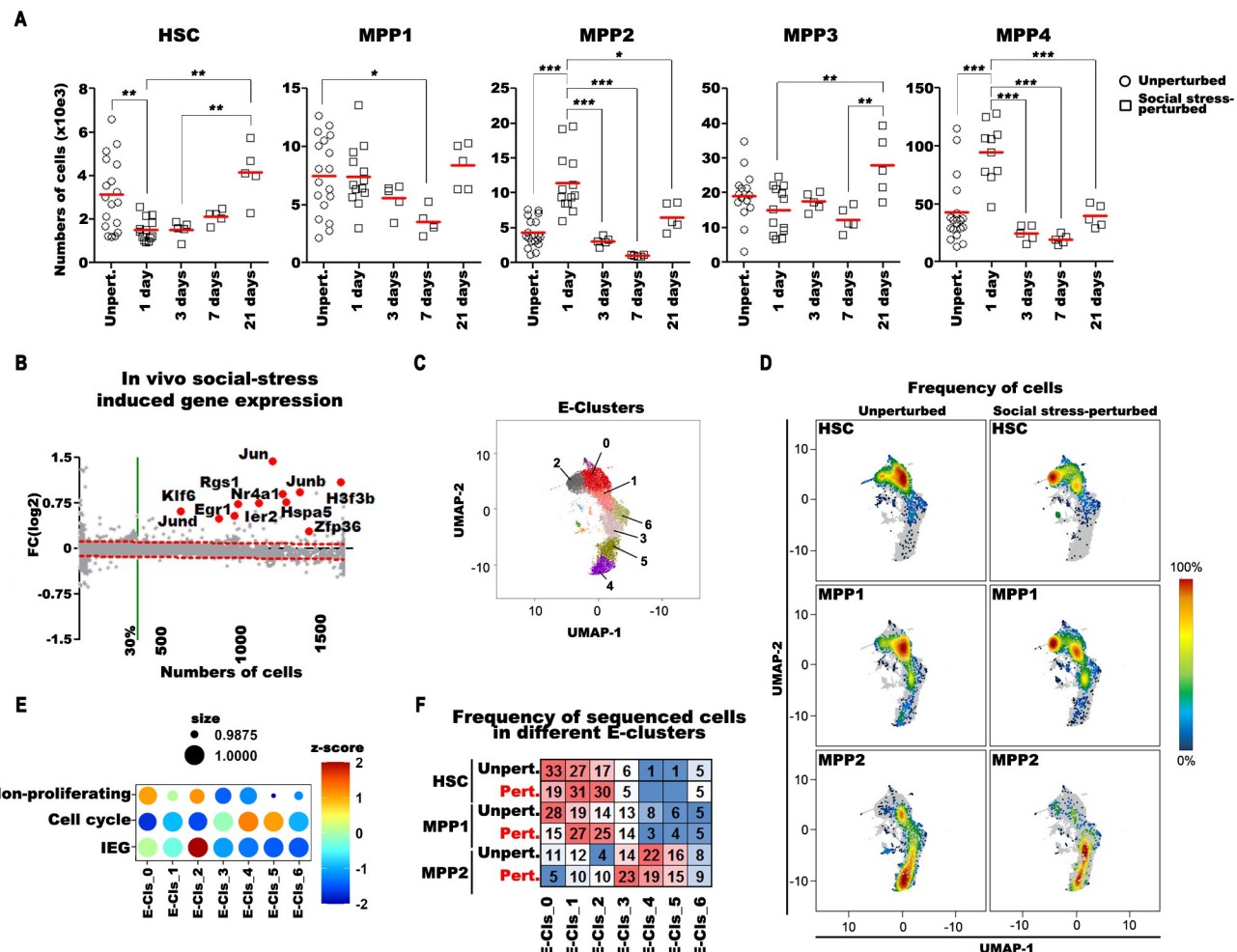

**Fig 1. The effect of in vivo social stress perturbation on hematopoietic cells in BM.** (**A**) The analysis of flow cytometry measurements on BM HSCs MPP1, 2, 3, and 4 cells without (unperturbed) and 1 day, 3 days, 7 days, or 21 days after short-term in vivo social stress perturbation. Single-cell suspensions from 2 tibia and 2 femurs/mice of unperturbed (Unpert./open circle) or in vivo social stress–perturbed (open squares at different time points) C57B6/J mice were stained for flow cytometry, measured, and the total numbers of cells/mice were plotted. Red lines indicate the mean values. One-way ANOVA with Tukey post normalization test was used to evaluate statistical significance (*, **, and *** indicate $p < 0.05$, $p < 0.01$, and $p < 0.001$, respectively. Data are available in S1 Data and on FlowRepository.org through FR-FCM-Z6PS accession number). (**B**) Genes with higher expression after short-term in vivo social stress perturbation (red dots and gene symbols are selected genes) in HSC are presented by comparative differential expression analysis of unperturbed versus perturbed cells. Differentially expressed genes are above the significance limit. The log2 fold-change expression values were plotted against the numbers of unperturbed cells express the gene. (**C**) The transcriptomes of single HSC, MPP1, and MPP2 cells of unperturbed or social stress–perturbed mice were analyzed by droplet-based scRNA sequencing. The samples were collected maximum 1 day after social stress induction. The clustering of early (**E**) hematopoietic compartment are referred as E-Clusters 0–6. (**D**) The frequency distribution (red to blue) of unperturbed and social stress–perturbed HSC, MPP1, and MPP2 cells were projected to UMAPs in order to show transcriptional program changes induced by social stress. (**E**) Expression of gene-set modules ("Non-proliferating," "Cell cycle," and "IEG") and coupled frequencies of cells expressing the gene-set in the different clusters are presented as Bubble-plots. (**F**) The distribution of sorted unperturbed (unpert.) and short-time in vivo social stress–perturbed (pert.) HSC, MPP1, and MPP2 cells is shown in the aggregated E-clusters. The frequency of a given cell type in E-clusters is colored from highest to lowest respective red to blue. The numerical data for Fig 1B–1F can be found in S2 Data. BM, bone marrow; HSC, hematopoietic stem cell; IEG, immediate early gene; MPP, multipotent progenitor; UMAP, Uniform Manifold Approximation and Projection for Dimension Reduction.

state to the social stress–perturbed state of HSC, MPP1, and MPP2 cells. All sequencing data were analyzed at a similar depth, displaying comparable number of genes and mRNAs per cell, allowing quantitative differential gene expression analyses.

We defined differentially expressed genes as expressed by (i) more than 30% of all unperturbed cells, (ii) with a significantly higher log2 fold-change between unperturbed and "in

vivo" social stress–perturbed cells, when adjusted to the 99% prediction band of a linear regression model. Genes above the noise level (red-dashed lines in **Figs 1B and S2**) are considered as significantly higher expressed in social stress–perturbed cells.

Comparison of gene expression levels in unperturbed and in socially stressed, perturbed HSC (**Fig 1B**), MPP1, or MPP2 (**S2 Fig**) 1 day after stress induction detected a perturbation-dependent general increase and an even more pronounced increase in the levels of transcription of IEGs beside a small set of approximately 90 genes in these cells (**Figs 1B and S2**). This suggests that perturbation of hematopoiesis by "in vivo" stress might use IEG-controlled activation.

## Comparison of whole transcriptomes of unperturbed with socially perturbed HSC, MPP1, and MPP2 detects differences in clusters of gene expressions

In primary cluster analysis of the transcriptomes of unperturbed and socially perturbed HSC, MPP1, and MPP2, 7 E-clusters (E for early hematopoiesis) were identified (**Fig 1C**), and the cell frequency distributions were projected in UMAPs (**Fig 1D**). E-0 cluster represents non-proliferating cells (mainly from HSC), E-1 activated toward cell cycle, and E-2 activated toward IEG expression (**Fig 1E**). By location on the UMAP, E-2 is not directed toward proliferating clusters, but its position may indicate an alternative activation. E-3-6 contain MPP1 and MPP2 expressing G1/S and G2/M cell cycle genes (**Figs 1E and S3A and S2 Data**).

Compared with unperturbed cells, social stress–perturbed cells in E-clusters were 14% less in nonproliferating E-0, 4% more in proliferation-activated E-1, 9% more in E-3, and 13% more in E-2 (**Fig 1F**).

The E-2 cluster contained IEG signatures (**Fig 1E**). Since its position within the UMAP landscape of clusters of transcriptionally relates cells is outside the G1/S/G2 cell cycle–active clusters E1 to E6, they may be differently activated. (**Fig 1D and 1E**). These results show that early progenitors are activated from quiescence by the stimulatory activity of short-term perturbation.

## The miR-221/222 gene cluster is expressed and can be deleted in all HSC and MPPs

The miR-221/222 gene cluster is located on the X chromosome and is expressed in HSC, in activated multipotent progenitors MPP1 and in proliferating MPP2, in myeloid-lymphoid-directed MPP3 and MPP4, and in CLP, granulocytes, and myeloid cells. We found that miR-221/222 cluster is turned off in BM preB cells, in thymocytes, and in peripheral blood derived, mostly nonactivated natural killer (NK) cells T and B lymphocytes (**S4A Fig**). These results support previous findings that resting cells do not express the miRNA family, but Th17 peripheral T cell subpopulations increase miR-221/222 expression upon activation [41]. It has been also described that miR-221/222 increased expression plays an important role in activated B cell Ig class switch in germinal center reactions [42].

We determined how many single HSC, MPP1, and MPP2 express how many miR-221 and miR-222 molecules per cell, by a standard curve-based, multiplexed stem-loop RT-TaqMan qPCR assay. We found that every tested HSC, MPP1, and MPP2 cell expressed between 7,000 and 8,000 copies of miR-221. Expression of miR-222, again tested in every cell, was lower and more variable between 1,000 and 5,000 copies per cell (**Fig 2A**). The level of miR-221 expression strongly correlated with miR-222 expression (**Fig 2B**).

To study functions of miR-221/222 during hematopoiesis, we generated mice, in which the one miR-221/222 cluster on the X chromosome in male mice could be deleted in all

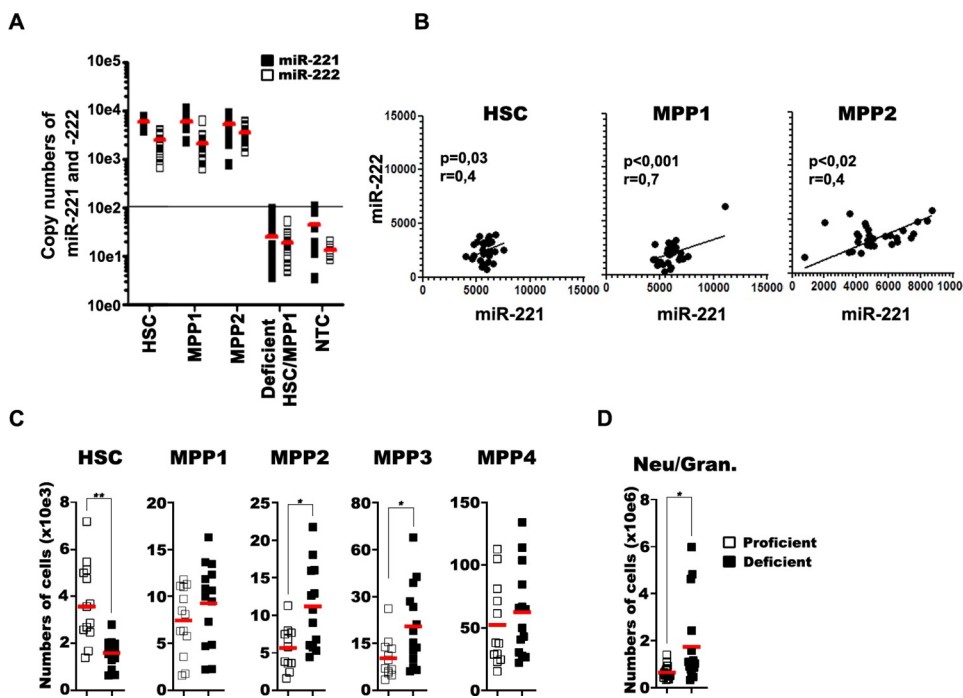

**Fig 2. Expression analysis of miR-221/222 in single cells and the effect of miRNA deficiency on hematopoietic stem cell populations in BM.** (**A**) Copy numbers of miR-221 (closed square) and miR-222 (open square) measured in single HSC, MPP1, and MPP2 of miRNA-proficient and in pools of HSC+MPP1 single cells of miRNA-deficient mice ($n_{cell}$ = 36 cells from 3 mice, $n_{NTC}$ = 12 wells). The highest value of NTC is marked as straight line, indicating the detection limit. (**B**) Correlation analysis of miR-221 and miR-222 copy numbers in proficient HSC, MPP1, and MPP2 cells (**C, D**) The analysis of flow cytometry measurements on BM and spleen derived cells. (**C**) Single-cell suspensions from 2 tibia and 2 femurs/mice or (**D**) from spleens of miR-221/222-proficient (open square) or miR-221/222-deficient (closed square) mice were stained for flow cytometry, measured, and the total numbers of cells/mice were plotted. Red lines indicate the mean values. Student *t* test was used to evaluate statistical significance (* or ** indicate $p < 0.05$ or $p < 0.01$, respectively). The numerical data for the graphs can be found in S1 Data and on FlowRepository.org through FR-FCM-Z6PS accession number). BM, bone marrow; HSC, hematopoietic stem cell; miRNA, microRNA; MPP, multipotent progenitor; NTC, no-template control.

hematopoietic cells. Vav[i-cre], active in all hematopoietic cells including HSCs, is commonly used to drive Cre recombinase expression in hematopoietic cells [4,40]. Thus, Vav-Cre mice were crossed with miR-221/222[fl/fl] mice to generate male F1 progeny, where the miR-221/222 cluster was expected to be deleted on the one X chromosome. All mouse strains had been back-crossed on C57BL/6J for at least 8 generations to ascertain genetic homogeneity, except for the miR-221/222 cluster in all offsprings. Bulk-sorted BM-lineage (lin)⁻c-kit⁺Sca1⁺ (LSK), and MPP cells (LSK CD150⁻CD48⁺) (**S4B Fig**), and single LSK CD150⁺CD48⁻ (pool of HSC and MPP1) cells of miR-221/222[fl/y-Tg(Vav1-icre)] mice were tested for quantitative miR-221/222 expression at the single-cell level. Indeed, our analyses show high efficiency of deletion of the miR-221/222 cluster in all tested cell populations. (**Figs 2A and S4B**).

## MiR-221/222 deficiency decreases HSC and increases MPP in unperturbed hematopoiesis

In BM of miR-221/222-deficient mice at steady state of unperturbed hematopoiesis, we found numbers of HSC reduced approximately 3-fold, whereas MPP2 and MPP3 were significantly increased. (**Fig 2C**). The extent of these changes in the numbers of HSC, MPP1, and MPP2 are similar to those seen in socially perturbed HSC, MPP1, and MPP2. This could suggest that

miR-221/222 deficiency and social stress might use, at least in parts, the same mechanisms to regulate pool sizes of early progenitors.

Numbers of MPP1, MPP4 (**Fig 2C**), CLP, CMP (**S1B Fig**) and MEPs and GMPs (**Fig 5B**) and peripheral CD4+, CD8+ T cells, and CD19+ B cells in spleen were not changed (**S1C Fig**). Interestingly, in spleen, the numbers of granulocytes were 6-fold elevated in miR-221/222-deficient mice (**Fig 2D; data also on S1C Fig**). These results suggest that expression of miR-221/222 maintains pool sizes of HSC and MPPs and prevents an increased granulopoiesis.

## MiR-221/222 targets *Fos* to protect the size of the unperturbed HSC pool

Next, we compared transcriptomes of deficient with proficient cells to detect microRNA-sensitive genes. We defined these genes as expressed by (i) more than 30% of all miR-221/222-proficien cells, (ii) with a significantly higher log2 fold-change between proficient and miR-221/222-deficient cells, when adjusted to the 99% prediction band of a linear regression model. Genes above the noise level (red-dashed lines in **Fig 3A**, **3E and 3F**) are considered as significantly expressed in miR-221/222-deficient cells.

At steady state of unperturbed hematopoiesis, we detected 9 such differentially expressed genes (**Fig 3A and 3B**). Among them, *Fos* was the only miR-221/222-seed-sequence-containing target gene (DIANA-TarBase v8 [43], TargetScan Release 8.0 [44], miRDB [45]), which has been experimentally validated by Errico and colleagues [46]. We validated this increased *Fos* expression with qPCR analyses (**Fig 3C**), in which *Fos* was found to be expressed 8.5-fold higher in deficient HSC. Six of the other 8 genes (*Jun*, *JunB*, *Nr4a1*, *Ier2*, *Zfp36*, and *Klf6*) are like *Fos* IEG [47,48] (**Fig 3A and 3D**).

Next, we performed the same analyses with MPP1 and MPP2. In unperturbed MPP1, *Fos*, *JunB*, *Nr4a1*, *Cd74*, *Ly6a*, *Zfp36*, and *Klf6* were no longer detectable, and *Rgs1* appeared (**Fig 3D and 3E**). In MPP2, no differentially expressed genes remained detectable (**Fig 3F**). This suggests that, in unperturbed hematopoiesis, up-regulated mRNA expression of *Fos*, *Jun*, and 5 other IEG transcription factors act primarily in HSC, causing a decrease of HSC and an increase of MPPs and of granulocytes.

We have also analyzed with gene set enrichment analysis (GSEA; [49,50]) if the expression of predicted miR-221/222 target genes (**Fig 4A**) have differential distribution in miR-221/222-deficient HSC, MPP1, and MPP2 cells according to E-clusters. These analyses show that the target gene expression is almost equally distributed without significant accumulation in any E-cluster between HSC, MPP1, and MPP2 cells (**Fig 4B and S2 Data**). We also could not detect significant difference in the cumulative expression between miR 221/222-proficient and miR 221/222-deficient cells as shown in **Fig 4C** and **S2 Data**. Additionally, we plotted the differential expression of each of the predicted target genes against the frequency of cells expressing the gene. We found 2-to-3 genes having marginal (approximately 5%) higher expression in at least 10% of deficient cells (**Fig 4D–4F**). Worth noting that 2-to-3 genes were also found to be higher expressed at the marginal level (approximately 5%) in miR-221/222-proficient cells, indicating a stochastic differential expression value around the 5% boundary. We found only Fos as experimentally validated direct target expressed approximately 20% higher in more than 60% of miR 221/222-deficient HSC and MPP1 cells.

## Shared IEG expression up-regulated by social stress perturbation and miR-221/222 deficiency

Next, we compared the differentially up-regulated genes in transcriptomes of social stress–perturbed and of experimentally unperturbed, miR-221/222 deficiency–influenced HSC, MPP1,

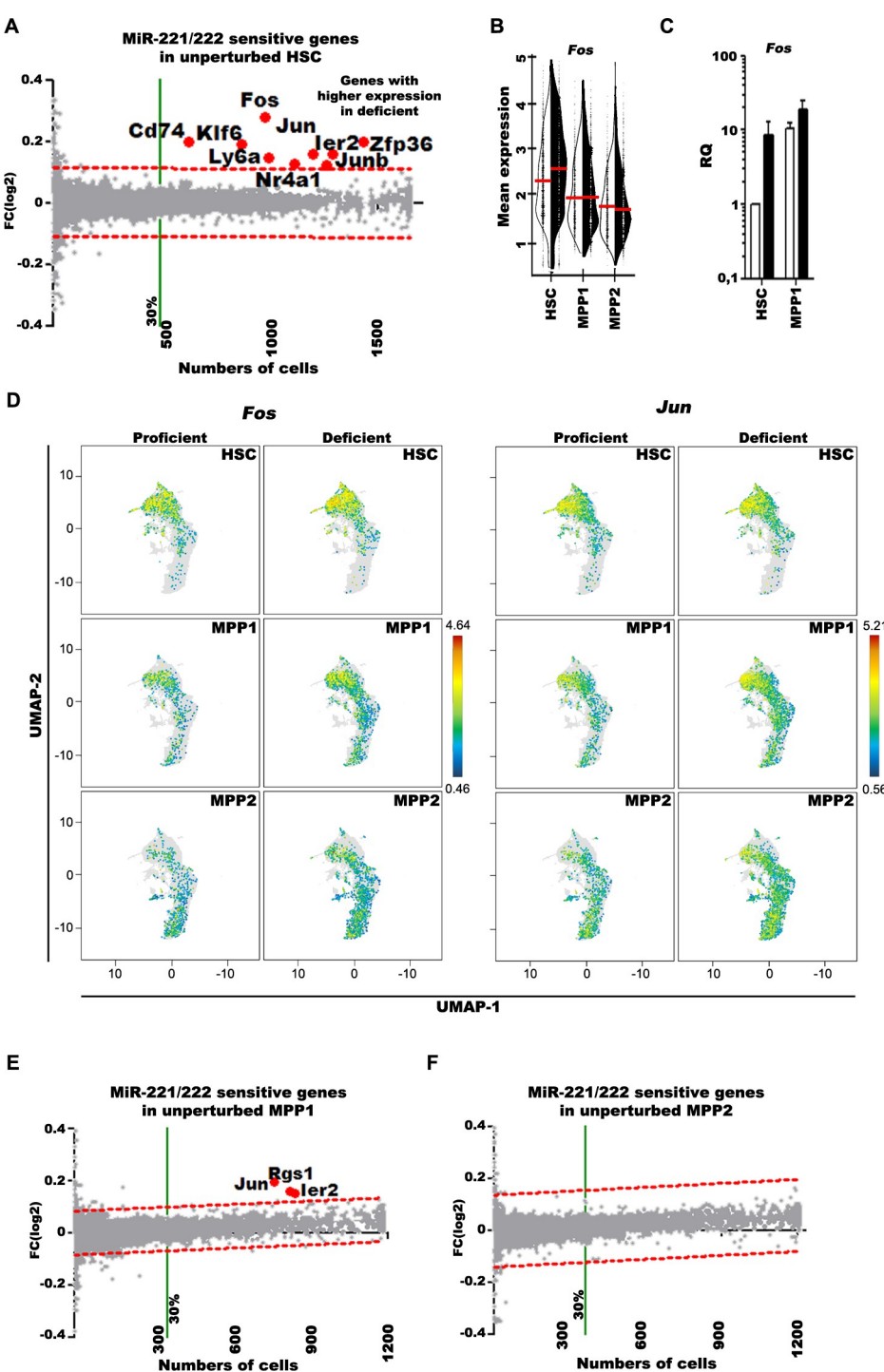

**Fig 3. miR-221/222-sensitive genes in unperturbed HSC, MPP1, and MPP2 cells.** (**A**) Comparative differential expression analysis between miR-221/222-deficient and proficient HSC, (**E**) MPP1, and (**F**) MPP2 cells. The log2 fold-change expression values are plotted against the numbers of proficient cells expressing the gene. After calculating the linear regression curve, the 99% CI band (dashed red lines) determined the nondifferentially (gray dots) and differentially (red dots and gene symbols) expressed genes. Significantly higher expressed genes in miR-221/222-deficient cells are above the 99% CI limit and are expressed by more than 30% of all cells (green line). These genes are called "differentially expressed genes." (**B**) Single-cell expression of *Fos* in HSC, MPP1, and MPP2 cells. The difference in the mean expression level of the indicated populations between proficient (open half violin) and deficient (filled half violin) cells are presented. (**C**) The RQ of *Fos* mRNA in bulk sorted miR-221/222-proficient (open bars) and miR-221/222-deficient (filled bar) HSC and MPP1 cells measured by RT-qPCR. RQ was calculated on the basis of *Hprt*

expression in miR-221/222-proficient HSC. (**D**) The differential expression of AP-1 components *Fos* and *Jun* presented on UMAP of unperturbed miR-221/222-proficient and miR-221/222-deficient HSC, MPP1, and MPP2 cells. The numerical data for **Fig 3A, 3B, and 3D–3F** can be found in **S2 Data**, and the numerical data for **Fig 3C** can be found in **S1 Data**. AP-1, activator protein-1; CI, confidence interval; HSC, hematopoietic stem cell; MPP, multipotent progenitor; RQ, relative quantity; RT-qPCR, quantitative reverse transcription PCR; UMAP, Uniform Manifold Approximation and Projection for Dimension Reduction.

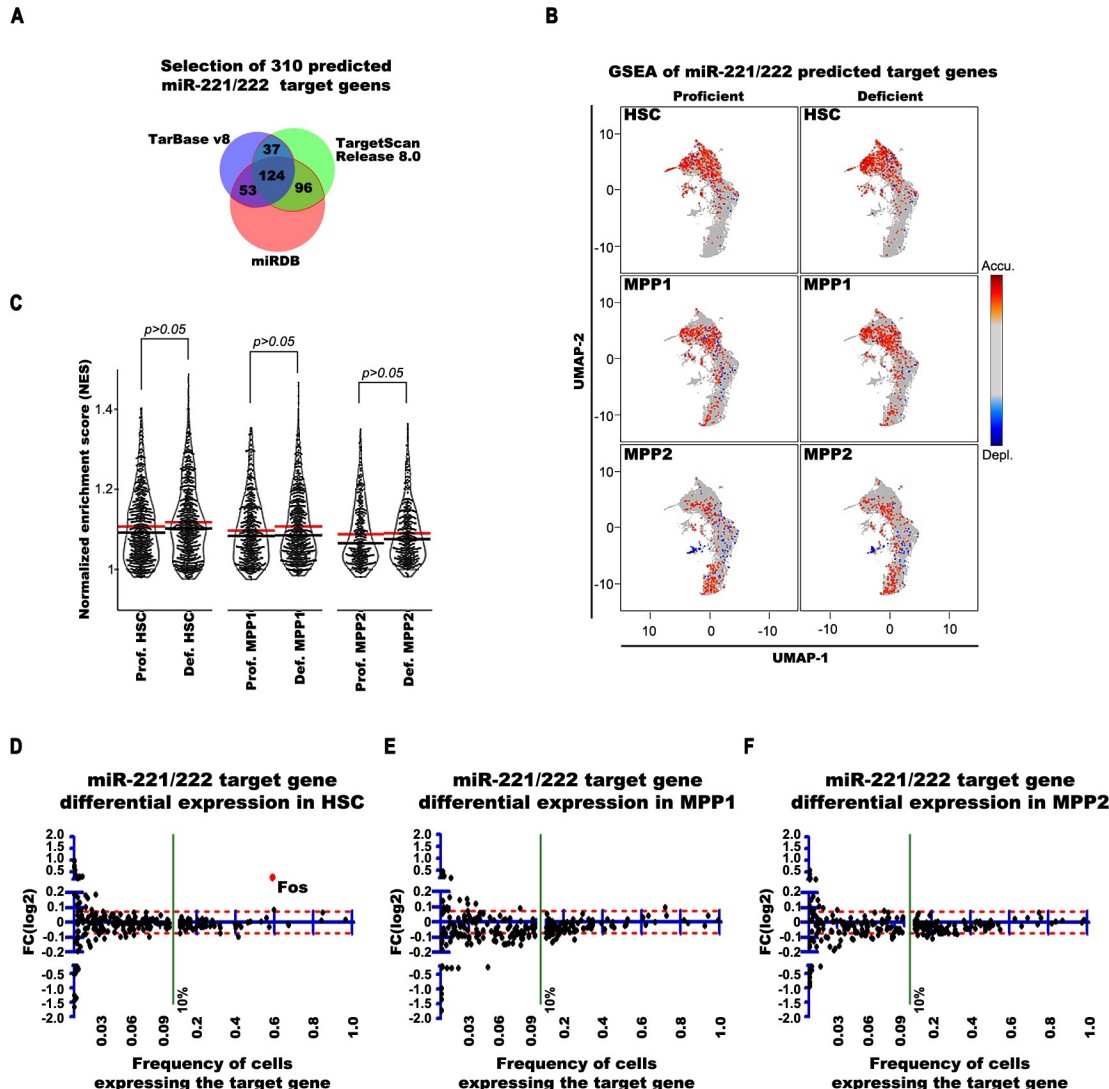

**Fig 4. GSEA and differential gene expression analysis of predicted miR 221/222 target genes in scRNA-seq data of miR-221/222-proficient and miR-221/222-deficient HSC, MPP1, and MPP2 cells.** (**A**) Selection of 310 predicted miR-221/222 target genes identified by at least 2 of 3 bioinformatical miRNA target prediction tools (TarBase v8, TargetScan Release 8.0 and miRDB). (**B**) GSEA was performed for each cell based on the difference to the mean of log normalized expression values of all cells in the analyzed set as preranked list and 1,000 randomizations. Significant up- or down-regulation was defined by an FDR ≤ 0.50 and normalized *p*-value < 0.05. For visualization, NES for significant cells were plotted. The GSEA was performed for indicated cells using the 310 predicted miR-221/222 target genes. (**C**) Violin plots show the mean (red bars) and median (black bars) differences of the NES of the GSEA in miR-221/222-proficeient and miR-221/222-deficient HSC, MPP1, and MPP2 cells. Significant differences between the NES values were calculated by Mann–Whitney U test. (**D-F**) Differential expression analyses of predicted miR-221/222 target gene expressions in miR-221/222-proficient and miR-221/222-deficient HSC (**D**), MPP1 (**E**), and MPP2 (**F**). The red dashed line indicates the 5% differential expression band. The numerical data for **Fig 4A–4F** can be found in **S2 Data**. FDR, false discover rate; GSEA, gene set enrichment analysis; HSC, hematopoietic stem cell; MPP, multipotent progenitor; NES, normalized expression score.

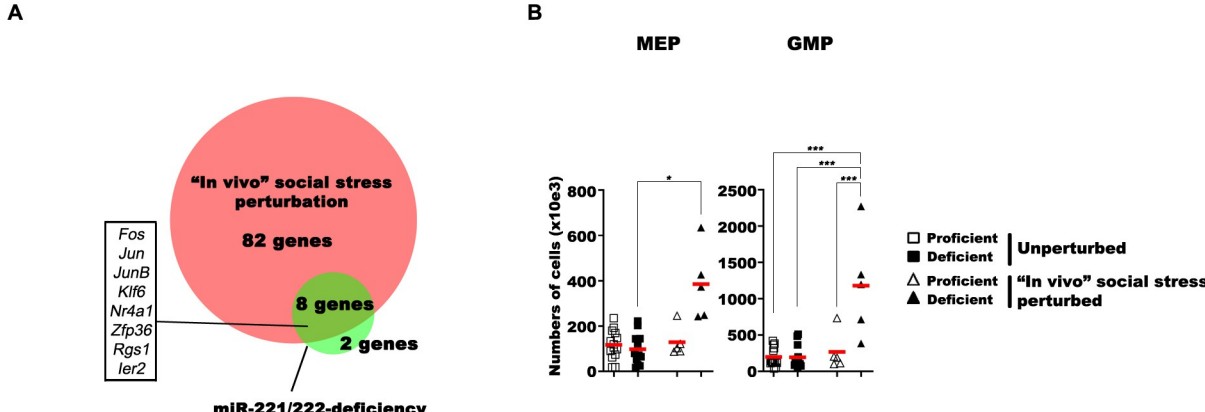

**Fig 5. Shared gene expression program in social stress–perturbed and miR-221/222 deficiency–induced HSC and MPP1 cells.** (**A**) The in vivo social stress–perturbed and miR-221/222 deficiency–induced genes were determined by differential gene expression analyses of scRNA-seq data and the plotted. The area-proportional Venn diagram shows the commonality between short-term social stress perturbation induced and miR-221/222 deficiency–sensitive genes. The full gene list is available from **S2 Data**. (**B**) Analysis of flow cytometry measurements on unperturbed and social stress–perturbed MEPs and GMPs. Single-cell suspensions of tibia and femurs of unperturbed miR-221/222-proficient (open squares) and miR-221/222-deficient (closed squares) mice or perturbed miR-221/222-proficient (open triangles) or miR-221/222-deficient (closed triangle) mice were prepared, analyzed with flow cytometry, and the numbers of cells were plotted. Red lines indicate the mean values. One-way ANOVA with Tukey posttest was used to evaluate statistical significance (*, **, and *** indicate $p < 0.05$, $p < 0.01$, and $p < 0.001$, respectively. Data are available in **S1 Data**, in **S2 Data**, and on [FlowRepository.org](FlowRepository.org) through FR-FCM-Z6PS accession number). GMP, granulocyte-myelocyte progenitor; HSC, hematopoietic stem cell; MEP, megakaryocyte-erythroid progenitor; MPP, multipotent progenitor.

and MPP2 cells. Six IEGs—*Fos*, *Jun*, *JunB*, *Ier2*, *Klf6*, and *Zfp36*–were found up-regulated by both stressors (**Fig 5A**). This suggests that HSC and MPPs use IEG responses to control pool sizes of HSC and their granulocyte biased differentiation. The results of our experiments indicate that this pathway is controlled by the miR-221/222 gene cluster.

## Controlled perturbation by social stress of miR-221/222-deficient BM leads to increases in numbers of MEP and GMP granulocyte progenitors

In unperturbed, miR-221/222-deficient mice increased numbers of granulocytes had been observed as a possible involvement of this miRNA cluster in enhanced granulopoiesis (**Fig 2D**). In support of this notion, short social stress (20 hours) was sufficient to induce increases in the numbers of MEP and GMP (in average 4-fold each) in miR-221/222-deficient, but not in miR-221/222-proficient BM (**Fig 5B**). The number of CLP and CMP was not different (**S1A and S1B Fig**). These rapid changes in granulocyte progenitor compartment sizes suggest that miR-221/222-deficient, stressed HSC and MPPs need only a short period of perturbation, in which proliferation and differentiation increase granulocyte-biased, but not lymphoid-directed hematopoiesis, detectable in increased numbers of MEP and GMP. By contrast, stress-induced changes in HSC and MPP numbers are not further altered by miRNA deficiency. Due probably to the short period of activation, the numbers of mature granulocytes in the periphery are not (yet) further increased (**S1B Fig**). We conclude from these results that miR-221/222 safeguards myeloid-lymphoid multipotency of HSC and MPP also in stressed hematopoiesis.

## Transcriptional landscapes of hematopoietic progenitor cells detect miR-221/222 deficiency–induced granulopoiesis

In order to construct comprehensive transcriptional landscapes of hematopoietic progenitors in BM at unperturbed, steady-state hematopoiesis, with trajectories of gene expressions during

development to different lymphoid, myeloid, and erythroid cell lineages and to detect possible changes in granulocyte-directed trajectories, we conducted separate transcriptome analyses for HSC, MPP1, and MPP2, including also more mature BM progenitors, i.e., MPP3, MPP4, CLP, and lin⁻c-kit⁺Sca1⁻ or lin⁻c-kit⁻Sca1⁻ cells from miR-221/222-proficient and miR-221/222-deficient BM. Cells were clustered (T-clusters for "total") according to their transcriptomes by Uniform Manifold Approximation and Projection for Dimension Reduction (UMAP) [51] (**Fig 6A**).

We selected sets of genes to characterize expression programs for HSC, cell -cycle, erythroid, megakaryocytic, lymphoid, myeloid, and granulocyte differentiation and assigned these T-clusters (**Figs 6B–6E** and **S3B** and **S2 Data**). These analyses are in agreement with earlier publications of other laboratories [1–8,52].

Only 2 T-clusters were different in proficient and deficient cells. T-7 (basophil/mast cell) contained −1.9-fold less, T-11 (granulocytes) 2.9-fold more cells in miR-221/222-deficient mice (**Fig 6E and 6F**). In fact, the granulocyte-related genes were only present in T-11 of miR-221/222-deficient, but not of proficient granulocytes (**Fig 6B and 6D**, lower left- and right-hand corners).

These results support our findings (**Fig 5B**) of increased numbers of granulocyte precursors in BM and granulocytes in the spleen miR-221/222-deficient mice. They suggest that steady-state hematopoiesis of miR-221/222-deficient BM progenitor cells increase granulopoiesis even in an experimentally unperturbed state.

## In perturbed HSC, MPP1, and MPP2 cells, miR-221/222 deficiency selectively increases transcription of genes encoding heat shock proteins, G protein–mediated tubulin, and chromatin remodeling activities

Finally, we compared transcriptomes of miR-221/222-deficient with proficient cells to detect microRNA-sensitive genes after "in vivo" social stress perturbation. We defined differentially expressed genes as expressed by (i) more than 30% of all in vivo social stress–perturbed miR-221/222-proficient cells, (ii) with a significantly higher log2 fold-change between in vivo social stress–perturbed miR-221/222-proficient and miR-221/222-deficient cells, when adjusted to the 99% prediction band of a linear regression model. Genes above the noise level (red-dashed lines in **Fig 7A–7C**) are considered as significantly expressed in in vivo social stress–perturbed miR-221/222-deficient cells.

Beside up-regulated IEG expression, social stress induction combined with miR-221/222 deficiency selectively up-regulated 3 groups of functionally connected genes in HSC, MPP1, and/or MPP2 (**Fig 7A–7C**):

1. Heat shock protein (HSP70) genes *Hspa5* and *Hspa8*, with possibilities to activate the unfolded protein response (UPR) [53].

2. G protein signaling (*Rgs1*) and its tubulin components (*Tubb1b*, *4b*, and *5*) in cytoskeleton and microtubule assembly, spindle formation, and mitotic cell cycle control [54]. Beyond the known activities of Rgs1 in hypoxia [55] and in SDF-1/CXCR4-induced migration of HSC [56], these genes are expected to contribute to increased erythroid-myeloid differentiation.

3. Replication-independent, histone-associated chromatin remodelers (*H3f3b*, *Hmgb2*, *H2afx*, *H2afz*). Since *H3f3b* is known to induce erythroid differentiation [57], miR-221/222-dependent up-regulation of *H3f3b* would favor erythroid-myeloid progenitor activation, as seen in our analyses.

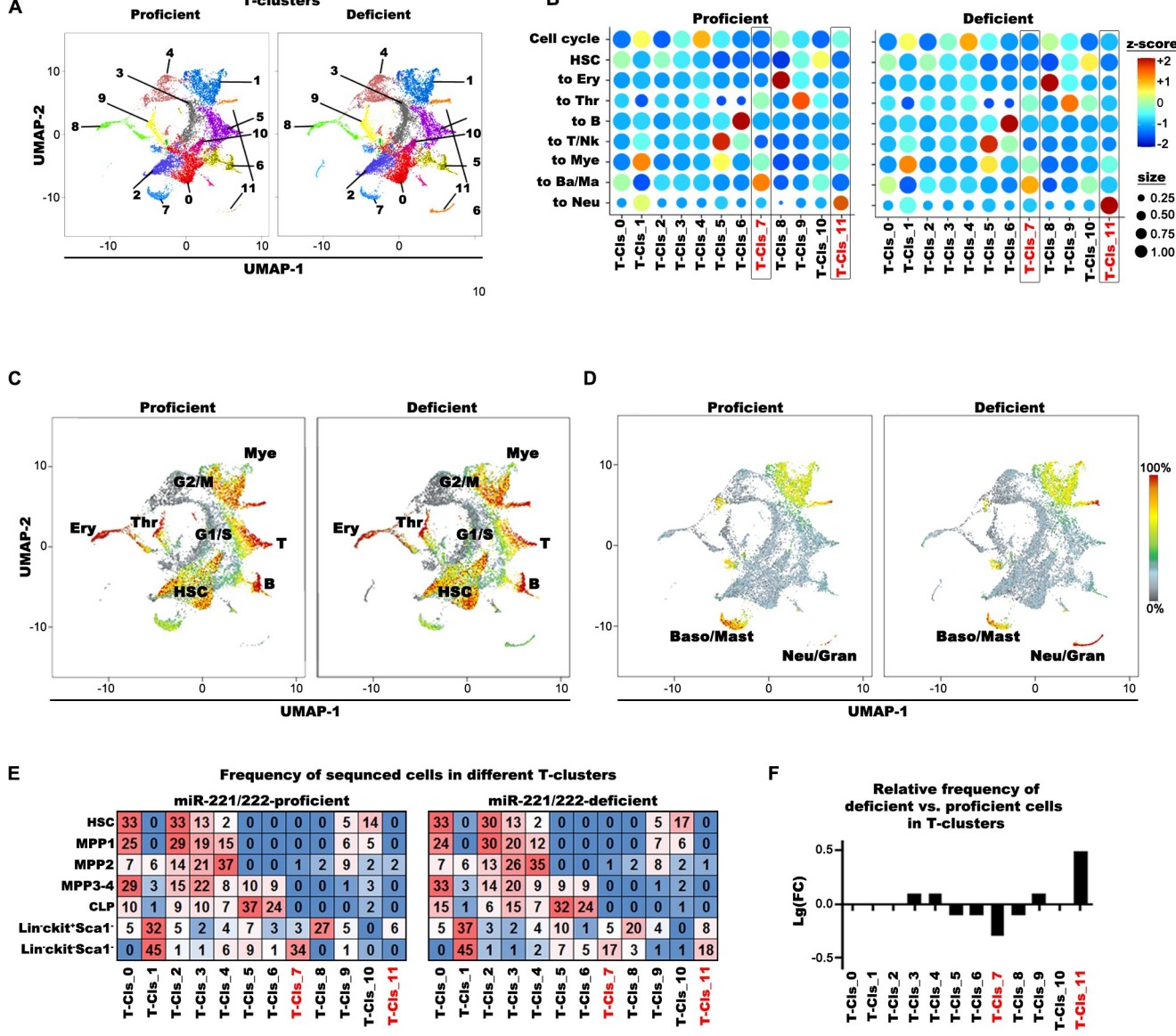

**Fig 6. Single-cell transcriptome analysis of unperturbed miR-221/222-proficient and miR-221/222-deficient lineage-negative BM compartment.** (**A**) Cluster analysis on the aggregated UMAP plots of miR-221/222-proficient and miR-221/222-deficient HSC, MPP1, MPP2, MPP3-4 pool, CLP, lin⁻ckit⁺Sca1⁻, and lin⁻ckit⁻Sca1⁻ populations (total, T-clusters) after single-cell transcriptome sequencing. Proficient and deficient cells are separately plotted. (**B**) The bubble-plots shows the expression of gene-set modules (z-score) and coupled frequencies of cells (bubble size) expressing the gene-set in the different T-clusters. miR-221/222-proficient and miR-221/222-deficient samples were separately plotted. (**C, D**) Major hematopoietic differentiation pathways are presented on the UMAP as characteristic gene-set modules (S2 Data) by the Log2 summary expression of the module genes. (**C**) Cells with characteristic expression pattern for Mye (to Myeloid), T (to T cells), B (to B cells), Ery (to erythrocytes), Thr (to thrombocytes) are shown on a composite picture for proficient and deficient cells and (**D**) for Ba/Ma (to basophil/mast cells) and Neu/Gran. (to neutrophils/granulocytes). (**E**) Frequencies of sorted miR-221/222-proficient and miR-221/222-deficient HSC, MPP1, MPP2, MPP3-4 pool, CLP, lin⁻ckit⁺Sca1⁻, and lin⁻ckit⁻Sca1⁻ cells in different T-clusters. The frequency of a given cell type in T-clusters is colored from highest to lowest respective red to blue. (**F**) Relative frequencies of cells in different T-clusters are presented on Log10 scale (frequency of deficient cells relative to the frequency of proficient cells). The numerical data can be found in S2 Data. BM, bone marrow; CLP, common lymphoid progenitor; HSC, hematopoietic stem cell; MPP, multipotent progenitor; UMAP, Uniform Manifold Approximation and Projection for Dimension Reduction.

We expect that the combined actions of all of these up-regulated genes could contribute to the activation of HSC from quiescence and to induce selectively increased granulopoiesis [52,58–61].

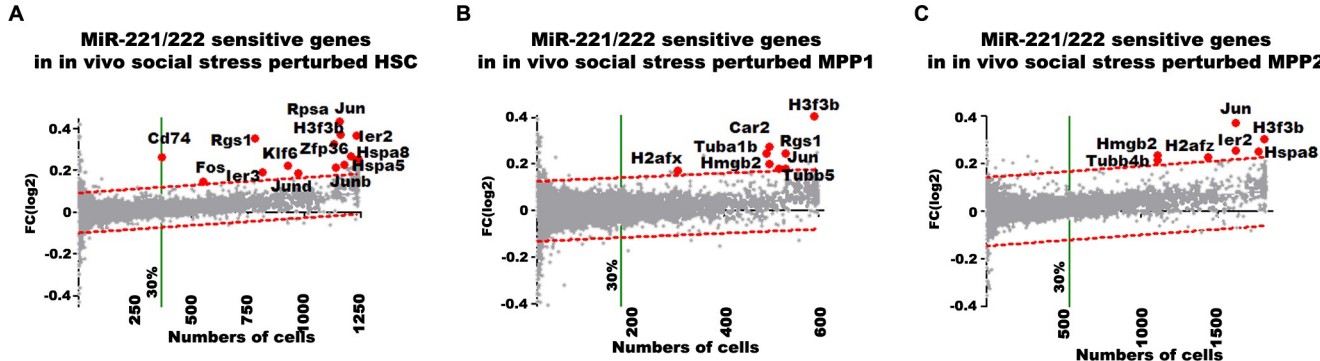

**Fig 7. Analysis of in vivo social stress perturbation–sensitive genes in miR-221/222-deficient HSC, MPP1, and MPP2 cells.** Genes with higher expression after short-term perturbation (red dots and gene symbols are selected genes) in miR-221/222-deficient (**A**) HSC, (**B**) MPP1, and (**C**) MPP2 are presented by comparative differential expression analysis of unperturbed versus perturbed cells. Differentially expressed genes are above the significance limit. The log2 fold-change expression values were plotted against the numbers of unperturbed cells express the gene. The numerical data can be found in **S2 Data**.

## miR-221/222-deficient HSC fail to reconstitute recipients in serial transplantations

In transplantations with the aim to reconstitute the hematopoietic cells of a lethally irradiated host, CD34⁻ HSCs have been found to contain long-term, fully repopulating HSC, while CD34⁺ MPP1 repopulate all lineages but have reduced capacities to reconstitute long-term repopulating HSC and early progenitors [1,4,5]. Serial transplantations of HSC can be expected to impose "in vitro" stress on HSC during their preparation for transplantations.

In order to test repopulation capacities of miR-221/222-proficient and of miR-221/222-deficient HSC, we transplanted 100 donor CD45.2⁺ either proficient or deficient HSC into lethally irradiated CD45.1⁺ mice, together with 10⁶ non-irradiated CD45.1⁺ BM carrier cells known to contain 100 proficient HSCs—conditions that are expected to lead to 50% CD45.1–50% CD45.2 chimerism. In serial transplantations, we then sorted CD45.2⁺ HSC from the transplanted mice and re-transplanted 100 HSCs under the same conditions.

In blood, the chimerism in total CD45⁺ hematopoietic cells was found to be close to these expected values after the first (approximately 40%) and second (approximately 55%) transplantations of miR-221/222-proficient HSC (**Fig 8A**). With miR-221/222-deficient HSC, this chimerism was significantly lower 4 months after the first (approximately 24%) and much lower 4 months after the second (2.5%) transplantations, suggesting that miR-221/222 deficiency exhausts the capacity of HSC for hematopoietic reconstitution.

Despite this reduced peripheral chimerism in deficient versus proficient T (15% versus 35%) and B cells (10% versus 30%), no differences in reconstitution were observed between proficient and deficient HSCs in BM 4 months after the second transplantation. Furthermore, secondary recipients of proficient HSC had 30% donor-derived myeloid cells and granulocytes, while recipients of deficient HSC contained 65%. This suggests that twice-transplanted miR-221/222-deficient HSC continued to develop granulocytes but had become deficient in their capacities to develop lymphoid cells (**Fig 8B**).

In BM, serial transplantations of proficient and deficient HSC generated different sizes of repopulated HSC and MPP2 pools (**Fig 8C**). We found an equal chimerism of proficient as well as deficient CD45.2⁺ HSC after the first transplantation, indicating comparable, adequate homing to the BM of both types of HSC. Four months after secondary transplantation, proficient HSC established normal HSC levels. Surprisingly, levels of twice transplanted deficient

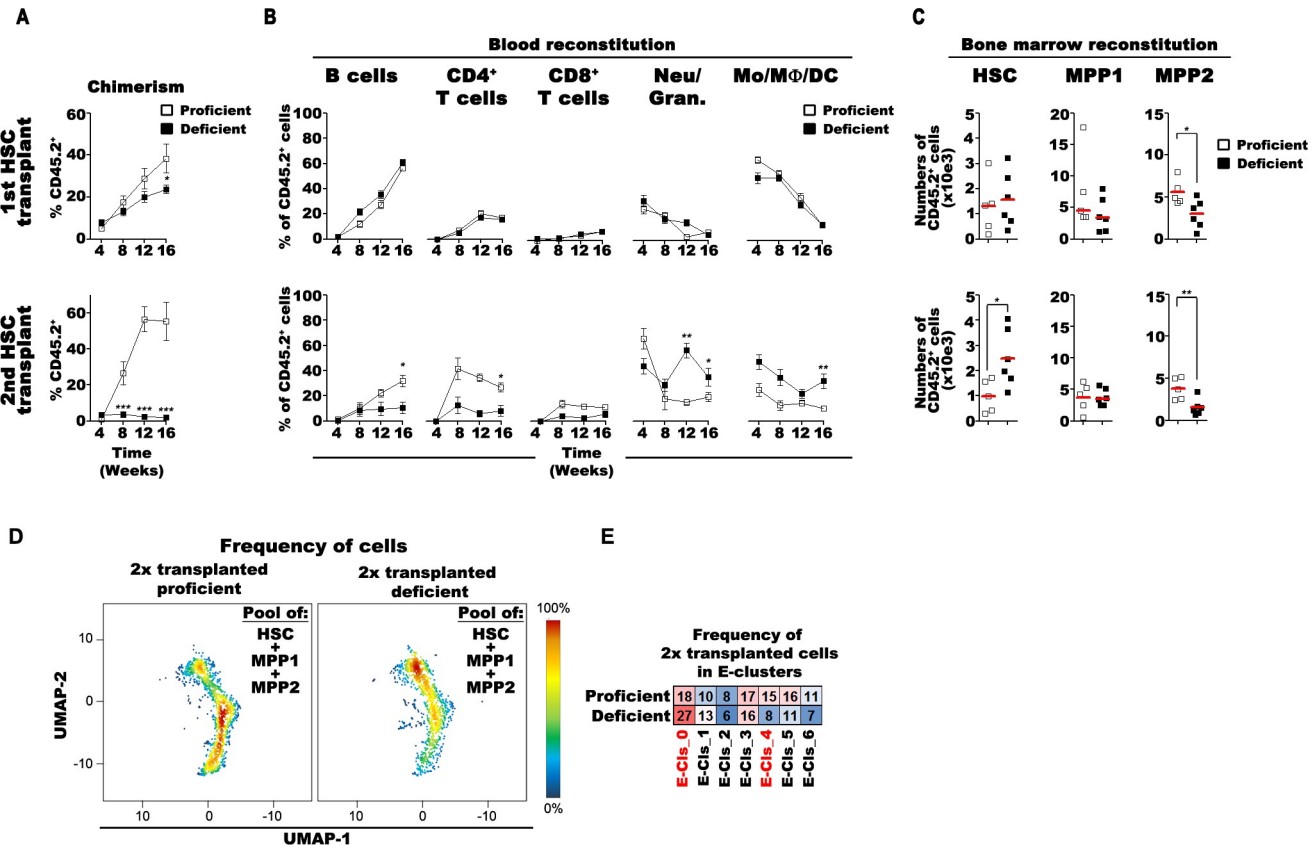

**Fig 8. miR-221/222 cluster safeguards the multipotency of HSC.** (**A**) Proportion of donor-derived blood cells after the first (upper) and second (lower) transplantation of 100 miR-221/222-proficient (open squares) and miR-221/222-deficient (closed squares) HSC. (**B**) Frequencies of B cells, CD4+/CD8+ T cells, CD11b+Gr1+ neutrophils/granulocytes (Neu/Gran.), and CD11b+Gr1− monocytes/macrophages/dendritic cells (Mo/MF/DC) of donor-derived blood cells after the first (upper line) or second (lower line) transplantation. (**C**) Numbers of donor-derived HSC, MPP1, and MPP2 cells 16 weeks after the first (upper line) or the second (lower line) miR-221/222-proficient (open squares) and miR-221/222-deficient (closed squares) HSC transplantation. Red lines indicate the mean values. (**A-C**) Data from 5 miR-221/222-proficient and from 6 deficient mice are presented as (**A, B**) mean values or (**C**) individually. Student *t* test was used to calculate significant differences (*, **, and *** indicate *p* < 0.05, *p* < 0.01, and *p* < 0.001, respectively. The numerical data for Fig 8A–8C can be found in S1 Data and on FlowRepository.org through FR-FCM-Z6PS accession number. (**D**) Density plots of pooled miR-221/222-proficient and miR-221/222-deficient HSC, MPP1, and MPP2 cells after serial transplantation are visualized on the aggregated UMAP and (**E**) the distribution of cells in the different E-clusters (see clusters in **Fig 1D**). The frequency of a given cell type in E-clusters is colored from highest to lowest respective red to blue. The numerical data for **Fig 8D and 8E** can be found in S2 Data. HSC, hematopoietic stem cell; MPP, multipotent progenitor; UMAP, Uniform Manifold Approximation and Projection for Dimension Reduction.

HSC levels were even 2- to 3-fold higher (**Fig 8C**). Numbers of MPP1 were not different, while MPP2 were reduced already after the first, and even more after the second transplantation of deficient HSC (**Fig 8C**). These data indicate that twice transplanted miR-221/222-deficient HSC retain their capacity to home to and repopulate their niches in BM but lose parts of their capacities of hematopoietic differentiation. Thus, the repeated "in vitro" stress during transplantation might favor the preservation, or even enhancement of granulocyte differentiation capacities of miR-221/222-deficient HSC, but it also extinguishes their lymphoid differentiation capabilities.

### Serial transplantations of miR-221/222-deficient HSC accumulate cell cycle–inactive HSC and deplete proliferating MPPs

Finally, we also compared twice serially transplanted miR-221/222-deficient HSC and MPPs, which had lost their multipotency but had retained their BM homing capacity (**Fig 8A–8C**)

with proficient, twice-transplanted progenitors. miR-221/222-deficient cells had increased numbers of E-0 (nonproliferating HSC) but decreased numbers of E-4-6 (cell cycle–active) (**Fig 8D and 8E**). This suggests that serial transplantation of miR-221/222-deficient HSC accumulates proliferation-inactive HSC and depletes cell cycle–active MPPs. These deficient HSC are reminiscent of differentiation-inactive HSC clones described by Pei and colleagues [8]. Again, we conclude from these results that miR-221/222 safeguards the multipotency of HSC.

## Discussion

From the time, when BM develops and is populated in special niches by long-lived HSC, steady-state hematopoiesis replenishes most central and peripheral hematopoietic cell compartments throughout life, with half-lives between a few days and a few weeks. Most HSC are generated once during embryonic development of the bone and its marrow to remain long-lived, quiescent cells throughout life [1–9]. A smaller part of these HSCs and more differentiated progenitors serve as life-long sources for this continuous regeneration. In mouse BM, a few thousand LSK CD150$^+$ CD48$^-$ CD34$^-$ HSC have the capacity to survive as quiescent cells without dividing much for years, even for life. When transplanted into recipient mice, a single HSC can repopulate the HSC compartment, all progenitor compartments, and all the mature hematopoietic lineages in normal numbers. Hence, HSC can find their niches in BM, can fill these niches by symmetric cell divisions, and can initiate multilineage differentiation to mature erythroid, megakaryocytic, myeloid, and lymphoid cells [1–9]. Our findings that serial transplantations of stressed, miR-221/222-deficient HSC do not alter this repopulation capacity, but affect hematopoietic multipotency, might be a useful system to study mechanisms controlling the homing and residence of HSC.

Perturbations of hematopoiesis by bacterial [15] or viral infections [13], by social stress [12], or by transplantation [1,4], mediated by interferon-α [18] or interferon-γ [19], by poly-I: C dsRNA [15], or by G-CSF [20] have all been found to impact on steady-state multilineage hematopoiesis of HSC and favor differentiation-restricted HSC, leading, e.g., to emergency granulopoiesis [21]. Such restrictions have been seen to become more prominent with aging [22,24]. Repeated, long-term perturbations of HSC during life, e.g., by infections [13], by repeated social stress [12], or during serial BM transplantations, as done in our experiments, all can contribute to HSC aging [33]. Our findings that social stress favors an alternative activation of HSC to increased granulopoiesis, that includes IEG activation, and that miR-221/222-deficiency activates further gene expressions might offer new ways to understand the choice of HSC to be activated either to steady-state multilineage differentiation or to stress-induced myelopoiesis and granulopoiesis.

Fos has been validated as direct target gene in cutaneous melanoma cells [46]. The very low numbers of HSC in a mouse, and the lack of tissue culture conditions to create larger numbers of cells, have not allowed us to validate Fos as miR-221/222-target in HSC. Bioinformatical tools predict at least 300 genes expressed in HSC, MPP1, and MPP2 cells to be miR-221/222 targets, suggesting that most of the predicted and validated targets are not susceptible to miR-221/222 action in these cells or regulated at the level of translation as it has been show with kit [38]. Again, even with Fos, not all unperturbed HSC or perturbed HSC show these changes (**Figs 1A** and **2C**). These restrictions in the action of the miR-221/222 cluster remain to be investigated in greater detail.

Three of the 6 IEGs selectively up-regulated in unperturbed hematopoiesis by miR-221/222 deficiency, namely, Klf6, Nr4a1, and Zfp36 all might cooperate to condition HSC for myeloid-granulocyte-biased hematopoiesis. Klf6, together with Runx1, promotes the transition of neutrophils from BM to blood [62,63].

Our finding that miR-221/222 selectively up-regulates the expression of Klf6 in HSC, that miR-221/222-deficient mice have increased levels of granulocytes in spleen, and that serial transplantation of miR-221/222-deficient HSC leads to the development of myeloid-granulocytic-biased HSC all may indicate that this promotion to neutrophil development begins in HSC in BM. Nr4a1, an orphan nuclear receptor, has been found expressed on myeloid-biased HSC [64]. Thus, miR-221/222 deficiency, promoting the up-regulated expression of Nr4a1 (**Fig 3A**), might cooperate with Klf6 to condition HSC for myeloid-granulocytic development. Zfp36, an AU-rich RNA-binding protein [65,66], has been found to suppress hypoxia and cell cycle signaling, activities that are known to influence HSC performances. In gene expression trajectories of neutrophil-granulocyte development from HSC, Zfp36 has been identified as a differentiation-determining factor [15,67,68]. It remains to be investigated how miR-221/222 deficiency focuses its effect in HSC to this selective set of genes, thereby conditioning them for myeloid-granulocytic development [67].

A deeper understanding of the molecular mechanisms, how HSCs regulate engraftment in BM and balance self-renewal versus differentiation, has important clinical implications for mobilizing defense against infections, for improving BM transplantations, and for fighting cancer of hematopoietic precursor cells. Our findings suggest that expression of the miR-221/222 cluster safeguards long-lived, quiescent HSC from Fos/AP-1-induced activation resulting in the reduction of HSC, the proliferation of MPPs, and their differentiation and myeloid-granulocytic hematopoiesis [9,21,24,69–71]. Protocols, which aim at activating miR-221/222 expression in HSC, may be helpful to maintain or reestablish hematopoietic potency through longevity, BM homing capacity, quiescence, and myeloid-lymphoid multipotency over myeloid-biased hematopoiesis.

## Materials and methods

### Ethics statement

All of the experimental procedures complied with the "National Regulations for the Care and Use of Laboratory Animals" approved by the Landesamt für Gesundheit und Soziales, Berlin (T0334/13, G0050-17).

### Mice

All mice were bred and kept in the breeding and experimental animal facilities of the Deutsches Rheumaforschungszentrum in Berlin Marienfelde and Berlin Mitte under SPF conditions. For all studies, 6 to 12 weeks old mice were used. miR-221/222$^{flox/flox}$ mice on C57B6J background in which miR-221 and 222 were flanked by loxP sites were generated in the Helmholtz Zentrum München, Deutsches Forschungszentrum für Gesundheit und Umwelt (MAH-2166). Heterozygous C57BL6.Cg-Commd10$^{Tg(Vav1-icre)A2Kio/}$J (The Jackson Laboratory) male mice were bred with miR-221/222$^{flox/flox}$ females to generate miR-221/222$^{fl/y- Tg(Vav1-icre)}$ miR-221/222 knockout males (deficient, Def.) in the F1 generation. Heterozygous C57BL6.Cg-Commd10$^{Tg(Vav1-icre)A2Kio/}$J male mice were used as controls (proficient, Prof.).

The Ly5.1 (CD45.1) male mice were obtained from Charles River and used in serial transplantation experiments. Mice were kept for at least 7 days in the experimental animal facility before they were taken in experiments or analysis.

### Social stress induction of mice

For short-term social stress–induced perturbation of the hematopoietic system, the mice were used within 1 day, 3 days, 7 days, or 21 days after the transport from the breeding facility in

Berlin-Marienfelde to the experimental animal facility Berlin-Mitte. Separation of the experimental animals in the breeding facility, transportation, and new housing in the experimental facility is a standardized protocol, and the separation, transportation, and adaptation to the new environment are certified stress factors for experimental mice.

### Bone marrow and spleen preparation

BM and spleen-derived single-cell suspensions were prepared as described earlier [72]. In order to minimize a potential continued "ex vivo" stimulation of gene expression, e.g., of IEG expression, which might occur during BM cell preparation, cell handling of the samples, which might influence the transcriptome analyses reported here (qPCR and single-cell RNAseq), femurs and tibia from killed mice were collected into tissue culture medium containing 2 μg/ml Actinomycin-D. Actinomycin-D is an efficient RNA polymerase inhibitor and is used to minimize the uncontrolled "ex vivo" activation of gene expression changes as also found in hematopoietic cells [73,74].

### Flow cytometry and cell sorting

We adhered to the guidelines for flow cytometry and cell sorting of hematopoietic stem and progenitor population and major lineage populations in the BM and the spleen as described earlier [72]. We used the nomenclature and gating strategy of BM-derived stem and progenitor populations described by [5–7]. All gating strategy, surface marker expression, and antibodies used in the studies are listed in **S5A–S5G Fig**.

### Preparation of total RNA

Total RNA was isolated from equal number of ex vivo sorted miR-221/222-proficient or miR-221/222-deficient cells using TRIzol Reagent (Invitrogen) according to the manufacturer's user guide. Isolated RNA was then quantified and qualified by Fragment Analyzer with the HS NGS Fragment Kit (1 to 6,000 bp) (Agilent).

### Real-time PCR analysis

RT-PCR was performed using SuperScript IV Reverse Transcriptase (Invitrogen) cDNA synthesis reaction and Oligo d(T)18 primer (Thermo Fisher Scientific) according to the user guide. Quantitive PCR was performed using QuantiTect SYBR Green PCR Master Mix (Qiagen) according to the user guide using following primer sets:

Fos F: 5′–GCCCAGTGAGGAATATCTGGA–3′, R: 5′–ATCGCAGATGAAGCTCTGGT–3′, and Hprt F: 5′– AAGCTTGCTGGTGAAAAGGA–3′, R: 5′– TTGCGCTCATCTTAGGCTTT–3′ as endogenous control.

miRNA expression on bulk sorted cells was measured by TaqMan real-time PCR. All reagents were obtained from Thermo Fisher Scientific. Measurements were analyzed using the Δ/ΔCT method relative to normalized miR-221 expression in pre-BI cells. Sno202 was used as a housekeeping gene. All qPCR reactions were performed in triplicates and originates from minimum 3 biological parallels.

### Serial transplantation

The mice were treated with 100 mg/l Baytril 1 week before and for 2 weeks after transplantation. For the first transplantations, pools of BM cells from 3 CD45.2[+] miR-221/222-proficient mice and from separate pools of BM from 3 CD45.2[+]-deficient mice, in which 100 HSC each were FACS sorted and transplanted into 11, respectively, 12 CD45.1[+] lethally irradiated (9.5

Gy) hosts (together with $10^6$ CD45.1$^+$ non-irradiated, radiation-lethality-protecting BM carrier cells). After 4 months, BM cells of 5 mice transplanted with proficient HSC and of 6 mice transplanted with deficient HSC were separately pooled, and the CD45.2$^+$ HSC FACS enriched. Again, in the second transplantation, 100 of these transplantation-derived HSC from mice transplanted with either proficient or deficient HSC were transplanted into 6 lethally irradiated CD45.1$^+$ hosts each (together with $10^6$ carrier cells). Their hematopoietic reconstitution potential was assessed after another 4 months. To follow the transplantation efficiency, 10 µl tail vain blood was collected in heparinized tubes, and the major lineage fractions were analyzed every 4 weeks until 16 weeks after transplantation by flow cytometry.

## Quantitation of miR-221/222 expression on the single-cell level

To measure the expression level of miR-221 and miR-222 in single HSC, MPP1, and MPP2 populations, we combined the protocols of [35,75,76] with slight modifications. Briefly, the single cells were sorted in lysis buffer suitable for the RT reaction and then 3-plex (miR-221, miR-222, and Sno202 specific) RT was carried out. Before the conventional simplex TaqMan qPCR, a 3-plexed cDNA preamplification PCR was done. We included external calibration standards consisting of 10-fold serial dilutions of synthetic miRNA oligonucleotides of $10^5$ to $10^0$ copies (S6A and S6B Fig). Analysis of these standards revealed that assay efficiency varied considerably and that sensitivity ranged between $10^0$ and $10^2$ molecules per reaction based on no template control (NTC) (S6C and S6D Fig). Pre-BI cells of Pax5$^{-/-}$ mice expressing high levels of miR-221 and miR-222 were used as positive control while setting up the protocol. Variations in assay efficiency were found independent of multiplexing level and were observed even for serial dilutions of miRNA standards in single-plex reactions; therefore, the Sno202 level of single-cell samples not reaching the NTC were considered as sorting failure and excluded from the analysis (<10% of all sorted cells).

## Single-cell RNA-library preparation and sequencing

Single-cell BM suspensions of unperturbed miR-221/222-proficient and miR-221/222-deficient mice were stained for preparative cell sorting and MPP, CLP, lin$^-$c-kit$^+$Sca1$^-$, and lin$^-$c-kit$^-$Sca1$^-$ (5,200 each) cells (S5 Fig) were applied to the 10x Genomics workflow for cell capturing and generation of scRNA gene expression (GEX) library using the Chromium Single Cell 3′ Library & Gel Bead Kit.

Due to the low cell counts of stem and progenitor cells, the BM-derived lineage-positive (B220$^+$, CD3$^+$, CD4$^+$, CD8$^+$, CD11b$^+$, CD11c$^+$,CD19$^+$, Gr1$^+$, NK1.1$^+$, TER119$^+$) cells were first depleted on magnetic column (LS from Miltenyi), and the remaining lin$^-$ single-cell suspensions from 4–4 perturbed or unperturbed miR-221/222-proficient and miR-221/222-deficient mice were labeled separately for preparative FACS (S5 Fig), followed by staining with TotalSeq-C anti-mouse Hashtag-1 or-4 (miR-221/222-proficient, perturbed: #1 or unperturbed: #4) or -Hashtag-2 or-5 (miR-221/222-deficient perturbed: #2 or unperturbed: #5) antibodies (BioLegend, #1: 155,861, #2: 155,863, #4: 155,867, #5: 155,869). The unperturbed or perturbed miR-221/222-proficient samples were pooled with deficient samples for scRNAseq. Then, 11,584 HSC, 12,605 MPP1, and 18,721 MPP2 sorted from pooled unperturbed samples or 8,900 HSC, 12,400 MPP1, and 9,100 MPP2 sorted from pooled perturbed samples were then applied to 10x Genomics workflow for cell capturing and scRNA gene expression (GEX) library preparation using the Chromium Single Cell 5′ Library & Gel Bead Kit as well as the Single Cell 5′ Feature Barcode Library Kit (10x Genomics). After cDNA amplification, the CiteSeq libraries were prepared separately using the Single Index Kit N Set A, while final GEX

libraries were obtained after fragmentation, adapter ligation, and final Index PCR using the Single Index Kit T Set A.

The BM-derived lineage-positive cells were first depleted on magnetic column, and the remaining single-cell suspensions from four 2-time transplanted animals were stained for preparative sorting of CD45.2$^+$ cells. After sorting 7,045 HSC, 6,363 MPP1, and 3,532 MPP2 from miR-221/222-proficent HSC transplanted mice, the cells were separately applied, while 1,780 HSC, 771 MPP1, and 422 MPP2 sorted from deficient HSC transplanted mice were pooled after sorting and applied to the 10x Genomics workflow using the Chromium Single Cell 5′ Library & Gel Bead Kit and Single Index Kit T Set A.

For all libraries prepared, the fragment sizes were determined using the Fragment Analyzer with the HS NGS Fragment Kit (1 to 6,000 bp) (Agilent), and library concentrations were determined with Qubit HS DNA assay kit (Life Technologies).

3′ GEX libraries and 5′ GEX+CITE libraries were sequenced on a NextSeq500 device (Illumina) using High Output v2 Kits (150 cycles) or on a NextSeq2000 device (Illumina) using either P2 reagents (200 cycles) or P3 reagents (200 cycles or 100 cycles) with the recommended sequencing conditions for (read1: 26 nt; read2: 98 nt; index1: 8 nt; index2: n.a.).

## Single-cell transcriptome profiling

Raw signals were demultiplexed and converted to fastq files using DRAGEN (Ilumina). Detection of intact cells and expression quantification was performed by cellranger (version 5.0.0) using count in default parameter settings with number of expected cells set to 3,000 and refdata-cellranger-mm10 as reference. Further analysis was done in R (version 4.1.2) using the Seurat package (version 4.0.5).

The integration of the datasets from HSC, MPP1, MPP2, MPP [3–4], CLP, lin$^-$c-kit$^+$Sca1$^-$, and lin$^-$c-kit$^-$Sca1$^-$ cells of unperturbed miR-221/222-proficient and miR-221/222-deficient mice was performed by following the integration pipeline as described in the FindIntegrationAnchors (Seurat) R Documentation. Firstly, each library was log-normalized using NormalizeData, 2,000 variable genes were detected using vst as selection method with FindVariableFeatures, variable genes were scaled using ScaleData, and 50 principle components were computed using RunPCA. Next, common anchors were identified by FindIntegrationAnchors using rpca as reduction, 2,000 anchor features, and 1:30 dimensions. Finally, libraries were merged using IntegrateData.

The integrated data were further analyzed by a UMAP sequentially using ScaleData, RunPCA for 50 principle components, and RunUMAP using 1:30 principle component. Transcriptionally similar clusters (T-clusters) were identified by shared nearest neighbor (SNN) modularity optimization with FindNeighbors using 1:30 principle components and FindClusters with the resolutions of 0.2. The UMAP was evaluated by projection of scores for selected developmental stages, defined as the sum of log-normalized expression values.

In the aggregated dataset, we detected a total of 15 distinct clusters of BM lin$^-$ populations (called total- (T-) clusters). Excluding clusters representing less than 5% of all analyzed cells, we further investigated T-cluster 0 to 11. T-clusters 0, 2, and 10 could be characterized by the expression of HSC-related genes, mostly from sorted HSC. T-clusters 3 and 4 had accumulated expression of cell cycle–active genes—cluster 3 for G1/S-phase and cluster 4 for G2/M phase, mostly from sorted MPP1-4 cells. T-cluster 5 contained T and NK lymphoid directed, T-cluster 6 contained B cell directed cells, mostly from sorted CLP. T-cluster 8 contained erythropoiesis-directed, T-cluster 9 megakaryocyte-platelet-directed genes, and T-cluster 1 had accumulated genes of earlier phases of myelopoiesis, mostly from sorted lin$^-$c-kit$^+$Sca$^-$ or lin$^-$c-kit$^-$Sca$^-$ cells. Finally, T-cluster 7 expressed genes of more mature stages of basophil and

mast cell development, while in T-cluster 11, genes expressed in more mature stages of neutrophils and granulocytes were predominant. These T-clusters 7 and 11 consisted mostly sorted lin⁻c-kit⁺Sca⁻ or lin⁻c-kit⁻Sca⁻ cells.

The same workflow was used to merge HSC, MPP1, and MPP2 datasets. In particular, the HSC, MPP1, and MPP2 from unperturbed and perturbed miR-221/222-proficient and miR-221/222-deficient mice, HSC, MPP1, and MPP2 sorted after the second transplantation of miR-221/222-proficient HSC as well as the pool of HSC, MPP1, and MPP2 sorted after the second transplantation of miR-221/222-deficient HSC were integrated, a UMAP was computed, and transcriptional cluster were defined using the resolution of 0.6. Clusters (E-clusters) with low quality were identify by visual inspection of UMI counts, number of expressed, and percentage of mitochondrial genes. The contaminating cluster with erythrocytes contained cells with high Hbb expression and higher UMI counts and number of detected genes. Cells from low-quality clusters as well as the contaminating cluster with less than 5% contribution to all cells were removed from further analysis. Libraries with hashtag-labeled antibody stainings of miR-221/222-proficient and miR-221/222-deficient cells were demultiplex by contrasting arc-sinh transformed hashtag 1 or 4 and hashtag 2 or 5 in a scatterplot and manual gating.

Differential gene expression analysis and comparison of number of expressed genes and UMIs per cell were performed based on downsampled libraries. In analogy to DESeq2 library size normalization, proposed for bulk sequencing [77], pseudo bulk samples were created for each library after removal of low-quality and contaminating clusters by summing up the read counts for each gene. Next, the geometric mean was calculated for each gene expressed in at least 75% of the considered cells in each library and at least 75% of miR-221/222-proficient and miR-221/222-deficient cells in the pools. The library size factors were defined as the median ratio between the UMI counts of the library and the corresponding geometric mean normalized to the number of considered cells. Downsampling was performed by subsampling of UMIs with sampling rates defined as the ratios of the minimal and the respective size factor. Numbers of detected genes and UMIs were computed after resembling the gene counts. Gene expression represents log2p-transformed UMI counts. Differential expression analysis was performed based on subsampled but not normalized values. In particular, for each gene log2p transformed, all 0 values removed and a Mann–Whitney test was performed. Next, genes were ranked by the fraction of expressing proficient (for miR-221/222-sensitive genes) or unperturbed (for perturbation-sensitive genes) cells, and a linear regression was performed on the log2p fold change between miR-221/222-deficient and miR-221/222-proficient or fold change between perturbed and unperturbed samples (mean $\log2Exp_{deficient}$-mean $\log2Exp_{proficient}$ or mean $\log2Exp_{perturbed}$-mean $\log2Exp_{unperturbed}$). The linear regression was performed using Prism (Version 9.0). Genes were defined as differentially expressed, if found in at least 30% of the cells and the log2 fold change exceeding the 99% prediction band (PB).

## Code availability

The softwares used in this study are open source. Cellranger from 10xgenomics: (https://support.10xgenomics.com/single-cell-gene-expression/software/downloads/latest) Seurat packages 4.1.1: (https://cloud.r-project.org/web/packages/Seurat/index.html). Used scripts are available upon request. (https://CRAN.R-project.org/CRANlogo.png; https://cloud.r-project.org/web/packages/Seurat/index.html) CRAN-Package Seurat: (https://cloud.r-project.org/web/packages/Seurat/index.html), (cloud.r-project.org). A toolkit for quality control, analysis, and exploration of single-cell RNA sequencing data. "Seurat" aims to enable users to identify and interpret sources of heterogeneity from single-cell transcriptomic measurements and to integrate diverse types of single-cell data. See [78–81] for more details.

## Supporting information

**S1 Data. Collection of numerical values in metadata table.**
(XLSX)

**S2 Data. Collection of single cell sequencing metadata normalized values.**
(XLSX)

**S3 Data. Collection of R scripts for single-cell RNA-seq data analysis.**
(ZIP)

**S1 Fig. Flow cytometry analysis of unperturbed and in vivo social stress–perturbed miR-221/222-proficient and miR-221/222-deficient hematopoietic cells in BM and spleen. (A)** Single-cell suspensions of 2 tibia and femurs of miR-221/222-proficient unperturbed mice or of 1 day, 3 days, 7 days, or 21 days after social stress–perturbed mice or **(B)** of miR-221/222-proficient and miR-221/222-deficient unperturbed or of 1 day after social stress–perturbed mice or **(C)** from spleens of miR-221/222-proficient and miR-221/222-deficient unperturbed or of 1 day after social stress–perturbed mice were prepared in matched pairs, analyzed with flow cytometry, and the numbers of cells were plotted. Numbers of different cell populations from unperturbed miR-221/222-proficient (open squares) and miR-221/222-deficient (closed squares) mice or perturbed miR-221/222-proficient (open triangles) or miR-221/222-deficient (closed triangle) are presented. Red lines indicate the mean values. One-way ANOVA with Tukey posttest was used to evaluate statistical significance (* indicates $p < 0.05$. The numerical data can be found in S1 Data and on FlowRepository.org through FR-FCM-Z6PS accession number).
(TIF)

**S2 Fig. Differentially expressed genes upon in vivo social stress perturbation in MPP1 and MPP2 cells.** Genes with higher expression after short-term perturbation (red dots and gene symbols are selected genes) in **(A)** MPP1 and **(B)** MPP2 cells are presented by comparative differential expression analysis of unperturbed versus perturbed cells. Differentially expressed genes are above the significance limit (red dashed). The log2 fold-change expression values were plotted against the numbers of unperturbed cells expressing the gene. The right side of the green line indicates genes expressed in more than 30% of the cells. The numerical data can be found in S2 Data.
(TIF)

**S3 Fig. G1/S and G2/M cell cycle phases on UMAPs of early (E) and total (T) hematopoietic compartments. (A)** The integrated data of miR-221/222-proficient and miR-221/222-deficient unperturbed, short-term perturbed, and serial transplanted HSC, MPP1, and MPP2 populations after single-cell transcriptome sequencing (early) or **(B)** HSC, MPP1, MPP2, MPP, CLP, lin⁻ckit+Sca1⁻ and lin⁻ckit⁻Sca1⁻ populations (total) were further analyzed by a UMAP. Cells with characteristic gene expression pattern for G1/S (left) or G2/M (right) cell cycle–related genes are shown on a composite picture for proficient and for deficient cells. For plotting, gene-set modules of G1/S and G2/M genes were used. The numerical data can be found in S2 Data. CLP, common lymphoid progenitor; HSC, hematopoietic stem cell; MPP, multipotent progenitor; UMAP, Uniform Manifold Approximation and Projection for Dimension Reduction.
(TIF)

**S4 Fig. miR-221/222 expression in hematopoietic cells. (A)** Relative miR-221 expression in bulk sorted hematopoietic progenitor, B cell subsets, NK cells, granulocytes, and T cells in BM

and thymus ($n = 5$ mice). Expression of miR-221 in the different bulk sorted populations are displayed as fold change relative to miR-221 expression in pre-BI cells. (**B**) Relative miR-221 expression in BM of miRNA-proficient and miRNA-deficient LSK and MPP subsets ($n = 5$ mice). One-way ANOVA with Dunnett posttest was used to evaluate statistical significance (\*, \*\*, and \*\*\*, indicate $p < 0.05$, $p < 0.01$, and $p < 0.001$, respectively). HSC/MPP1: pool of hematopoietic stem cell and multipotent progenitor (MPP)1, MPP3-4: pool of MPP3 and MPP4 populations, CLP: common lymphoid progenitor, Pre-BI: precursor BI cell, Pro-Pre-B: progenitor of precursor B cell, NK cell: natural killer cells. LSK: BM-derived lin$^-$Kit$^+$Sca1$^+$ population. The numerical data for Fig 1B–1F can be found in S1 Data. BM, bone marrow; HSC, hematopoietic stem cell; miRNA, microRNA; MPP, multipotent progenitor; NK, natural killer.
(TIF)

**S5 Fig. Gating strategy and surface marker expression of stem and progenitor populations in the BM and lineage cells in the spleen.** After preparing single cell suspensions of (**A**) unperturbed or (**B**) perturbed miR-221/222-proficient or of (**C**) unperturbed or (**D**) perturbed miR-221/222-deficient BM, stem and progenitor cells were stained for flow cytometry analysis. (**E**) Single-cell suspension of miR-221/222-proficient or miR-221/222-deficient spleen or blood was prepared, and the major hematopoietic lineage cells were stained for flow cytometry analysis. (**F**) Surface marker expression of the measured hematopoietic stem, progenitor, and lineage cells. HSC were gated on lin$^-$
(B220$^-$CD3$^-$CD4$^-$CD8$^-$CD19$^-$CD11c$^-$CD11b$^-$Gr1$^-$NK1.1$^-$TER119$^-$) c-kit$^+$S-ca1$^+$Flk2$^-$CD34$^-$CD150$^+$CD48$^-$ cells. MPP1 was gated on lin$^-$c-kit$^+$S-ca1$^+$Flk2$^-$CD34$^+$CD150$^+$CD48$^-$ cells. MPP2 were gated on lin$^-$c-kit$^+$Sca1$^+$Flk2$^-$CD34$^+$CD150$^+$CD48$^+$ cells. MPP3 were gated on lin$^-$c-kit$^+$S-ca1$^+$Flk2$^-$CD34$^+$CD150$^-$CD48$^+$ cells. MPP4 were gated on lin$^-$c-kit$^+$S-ca1$^+$Flk2$^+$CD34$^+$CD150$^-$CD48$^+$ cells. CLPs were gated on lin$^-$c-kit$^{lo}$Sca1$^{lo}$Flk2$^+$IL7R$^+$ cells. CMPs were gated on lin$^-$c-kit$^+$Sca1$^-$CD34$^+$CD16/32$^-$ cells. MEPs were gated on lin$^-$c-kit$^+$S-ca1$^-$CD34$^-$CD16/32$^-$ cells. GMPs were gated on lin$^-$c-kit$^+$Sca1$^-$CD34$^+$CD16/32$^+$ cells. CD4 T cells were gated as CD3$^+$CD4$^+$; CD8 T cells were gated as CD3$^+$CD8$^+$ cell. B cells were gated as CD4$^-$CD8$^-$B220$^+$CD19$^+$ cell. Myeloid cells were gated as CD4$^-$CD8$^-$B220$^-$CD19$^-$CD11b$^+$Gr1$^-$ cells. Granulocytes (Gran.) were gated on CD4$^-$CD8$^-$B220$^-$CD19$^-$CD11b$^+$Gr1$^+$ cells. (**G**) Antibodies used in the analyses. The numerical data can be found in the FlowRepository.org through FR-FCM-Z6PS accession number. BM, bone marrow; CLP, common lymphoid progenitor; CMP, common myeloid progenitor; GMP, granulocyte-myelocyte progenitor; HSC, hematopoietic stem cell; MEP, megakaryocyte-erythroid progenitor; MPP, multipotent progenitor.
(TIF)

**S6 Fig. Strategy for the detection of copy numbers of miR-221/222 molecules in single cells.** (**A**) Serial dilution of $10^5$–$10^0$ copies of synthetic miR-221 oligonucleotide was measured 3 times (Run1-3) in 3 technical replicates. The TaqMan qPCR measurements were done after or without (without pre-amp) 10-cycle PRC amplification of the reverse transcript. (**B**) Serial dilution of $10^5$–$10^0$ copies of synthetic miR-222 oligonucleotides were measured 2 times (Run1 and 2) in 3 technical replicates. The TaqMan qPCR measurements were done after 10-cycle PRC amplification of the reverse transcript. (**C**) A total of 45 single-cell sorted miR-221/222-proficient lin$^-$c-kit$^+$Sca1$^+$CD150$^+$CD48$^-$ (pool of HSC and MPP1) cells were directly sorted in lysis buffer, suitable for miR-221- or miR-222-specific reverse transcription reaction. After 10-cycle PRC amplification of the reverse transcript, the copy numbers of miR-221 (red) and miR-222 (blue) were determined in individual cells using standard curves developed in

the same reaction plate. Red and blue dots indicate technical replicates (**D**) The limit of detection in miR-221 and -222 copy numbers were assigned by the highest value of 10 NTCs, where no cell was sorted in the lysis buffer. Red and blue dots indicate technical replicates. Pax5$^{-/-}$ pre-BI cells were used as positive control. The mean of the technical replicates are calculated and plotted for 10 individual cells. The numerical data can be found in S1 Data. HSC, hematopoietic stem cell; MPP, multipotent progenitor; NTC, no template control.
(TIF)

## Acknowledgments

We thank Andreas Radbruch, DRFZ Berlin, and Nikolaus Dietlein and Hans-Reimer Rodewald, DKFZ Heidelberg, for critical reading of our manuscript. We thank the members of the animal breeding and experimental facility (Deutsches Rheuma-Forschungszentrum (DRFZ)) for their technical support and advice; the staff at the Max Planck Institute for Infection Biology Flow Cytometry Core Facility for expertise and instrument support; V.D. Dang, L. Bauer, and K. Lehmann for technical support and advice. We thank the DRFZ's core services for excellent support in technology and expertise.

## Author Contributions

**Conceptualization:** Peter K. Jani, Georg Petkau, Pawel Durek, Mir-Farzin Mashreghi, Fritz Melchers.

**Data curation:** Peter K. Jani, Georg Petkau, Gabriela Maria Guerra, Gitta Anne Heinz, Frederik Heinrich, Pawel Durek.

**Formal analysis:** Peter K. Jani, Frederik Heinrich, Pawel Durek, Fritz Melchers.

**Funding acquisition:** Fritz Melchers.

**Investigation:** Peter K. Jani, Georg Petkau, Yohei Kawano, Pawel Durek, Mir-Farzin Mashreghi, Fritz Melchers.

**Methodology:** Peter K. Jani, Georg Petkau, Yohei Kawano, Uwe Klemm, Gabriela Maria Guerra, Gitta Anne Heinz, Frederik Heinrich, Pawel Durek, Mir-Farzin Mashreghi, Fritz Melchers.

**Project administration:** Peter K. Jani, Uwe Klemm, Gabriela Maria Guerra, Gitta Anne Heinz, Mir-Farzin Mashreghi.

**Resources:** Peter K. Jani, Uwe Klemm, Gabriela Maria Guerra, Gitta Anne Heinz, Mir-Farzin Mashreghi.

**Software:** Peter K. Jani, Frederik Heinrich, Pawel Durek, Mir-Farzin Mashreghi.

**Supervision:** Peter K. Jani, Uwe Klemm, Pawel Durek, Fritz Melchers.

**Validation:** Peter K. Jani, Georg Petkau, Yohei Kawano, Uwe Klemm, Gabriela Maria Guerra, Gitta Anne Heinz, Pawel Durek, Fritz Melchers.

**Visualization:** Peter K. Jani, Frederik Heinrich, Pawel Durek, Fritz Melchers.

**Writing – original draft:** Peter K. Jani, Georg Petkau, Fritz Melchers.

**Writing – review & editing:** Peter K. Jani, Georg Petkau, Yohei Kawano, Uwe Klemm, Gabriela Maria Guerra, Gitta Anne Heinz, Frederik Heinrich, Pawel Durek, Mir-Farzin Mashreghi, Fritz Melchers.

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
