## [Editor Report · Decision Letter 0]

23 Jan 2023

Dear Dr Jani, 

Thank you for submitting your manuscript entitled "Titel MicroRNA-221/222-expression in HSC and MPP safeguards their quiescence and multipotency by downregulating stress-independent and dependent expression of IEG and of several myelo/granulopoiesis-enhancing target genes." for consideration as a Research Article by PLOS Biology.

Your manuscript has now been evaluated by the PLOS Biology editorial staff as well as by an academic editor with relevant expertise and I am writing to let you know that we would like to send your submission out for external peer review.

Once your full submission is complete, your paper will undergo a series of checks in preparation for peer review. After your manuscript has passed the checks it will be sent out for review. To provide the metadata for your submission, please Login to Editorial Manager (https://www.editorialmanager.com/pbiology) within two working days, i.e. by Jan 25 2023 11:59PM.

Kind regards,

Lucas

Lucas Smith, Ph.D.

Associate Editor

PLOS Biology

lsmith@plos.org

---

## [Decision Letter · Decision Letter 1]

21 Mar 2023

Dear Dr Jani,

Thank you for your patience while your manuscript entitled "Titel MicroRNA-221/222-expression in HSC and MPP safeguards their quiescence and multipotency by downregulating stress-independent and dependent expression of IEG and of several myelo/granulopoiesis-enhancing target genes" was peer-reviewed at PLOS Biology. Please also accept my apologies for the delay in providing you with our decision. The manuscript has now been evaluated by the PLOS Biology editors, an Academic Editor with relevant expertise, and by three independent reviewers. 

The reviews are attached below. As you will see, the reviewers find the conclusions interesting, but they also raise several issues that would have to be addressed to strengthen the results before we can consider the manuscript for publication. Reviewer 1 thinks you need to add further experiments to the ex vivo/in vitro stress section to confirm the conclusions and to identify direct targets of miR-221/222 among other issues. Reviewer 2 asks for several clarifications, including confirmation on which chromosome the miRNA 221/222 locus is located, and Reviewer 3 misses some controls and stats in specific experiments and agrees with the other reviewers that the manuscript would benefit from some rewriting to streamline the Introduction and Discussion.

In light of the reviews and discussions with the Academic Editor and the rest of the team, we would like to invite you to revise the work to thoroughly address the reviewers' reports.

Given the extent of revision needed, we cannot make a decision about publication until we have seen the revised manuscript and your response to the reviewers' comments. Your revised manuscript is likely to be sent for further evaluation by all or a subset of the reviewers.

**IMPORTANT - SUBMITTING YOUR REVISION**

3. Resubmission Checklist

a) *PLOS Data Policy*

b) *Published Peer Review*

d) *Blurb*

Please also provide a blurb which (if accepted) will be included in our weekly and monthly Electronic Table of Contents, sent out to readers of PLOS Biology, and may be used to promote your article in social media. The blurb should be about 30-40 words long and is subject to editorial changes. It should, without exaggeration, entice people to read your manuscript. It should not be redundant with the title and should not contain acronyms or abbreviations. For examples, view our author guidelines: https://journals.plos.org/plosbiology/s/revising-your-manuscript#loc-blurb

Sincerely,

Ines

--

Ines Alvarez-Garcia, PhD

Senior Editor

PLOS Biology

Reviewers' comments

Rev. 1:

Jani et al show that miR-221/222 deletion in hematopoietic stem cells (using Vav-iCre) affects the gene expression program of HSC and early multipotent progenitors in unperturbed and perturbed states (i.e. social disturbance), and that this results in changes in the number of precursors and mature granulocytes. Competitive serial transplantation revealed a substantial defect in miR-221/222 deficient cells in at least one set of experiments. The paper contains a distracting section about "ex vivo/in vitro stress" that is not important for the overall conclusions of the paper and may have a serious flaw in data interpretation. The handling of the direct effects of miR-221/222 on target mRNAs may reveal a lack of miRNA expertise on the research team. This part should be revisited. Nevertheless, the biological effects, if reproduced, are very interesting and worthy of report.

1. The first section of the results ("Controlled perturbations of steady-state hematopoiesis can be induced..) is redundant with a long section in the introduction, and references no new data or figure panels. One of these sections should be eliminated.

2. The "ex vivo/in vitro" stress results are distracting and unnecessary, especially because of one major caveat to their interpretation. The higher abundance of IEG genes in cells allowed to continue transcription during processing may reflect increased transcription due to activation of signaling pathways during processing, as the authors suggest. However, it is also possible that these genes are expressed at a constant basal rate in vivo and during processing, and that the observed differences in transcript abundance at the end of processing with or without ActD treatment reflects the very short half life of most of these mRNAs. That is, they are degraded during processing, and can't be replaced in the presence of ActD. Many of the highlighted IEGs are known to have very short-lived mRNAs, and ActD treatment of cells is the standard way to reveal this. This possibility is not mutually exclusive with transcriptional activation, but it is important to distinguish these effects. Is there really a transcriptional stress response going on during processing? Or is there a technical artefact in the gene expression analysis that is uncovered by ActD treatment? A more controlled in vitro stress would be preferable to this passive protocol that depends on ActD. It would also help to see evidence that stress pathways are induced (as in ref 11), assuming signaling state may be as easy to measure before and after processing. Another reasonable course of action would be to remove this section from the paper and just state that ActD was used during processing to avoid any effect of transcription occurring during cell processing.

3. Line 234-235: "[miR221/222] is turned off in natural killer (NK), thymocytes and T lymphocytes, and preB and B lymphocytes." This should more specifically reference the data in Supplementary Fig. 4A, which is focused on bone marrow and thymus. Some peripheral B and T cells do express miR-221/222, where it has been shown to function in immune responses.

4. The Methods section is inadequate in several sections, because the supplementary methods were not provided in the pdf package for reviewers. In particular, the single cell quantification of miRNA expression is not explained at all, though it seems quite novel and interesting. I see little need for supplementary methods, and recommend putting all relevant information in the main manuscript.

5. The insertion of figure legends within the Results section is distracting. Also, Figure 2 legend is repeated.

6. Lines 269-274: miR-221/222 is on the X chromosome, not the Y chromosome.

7. The direct targets of miR-221/222 would not be expected to have large individual changes. Finding one that was increased 8.5 fold (Fos) is not good experimental evidence of direct miRNA targeting of the Fos mRNA, but likely reflects secondary effects on Fos (like several other IEGs, as observed by the authors). These secondary effects are of interest(!), but they should be characterized as the emergent phenotypic effect of miR-221/222 deficiency, and potential direct targets should be assessed as well. To detect direct miR-221/222 silencing activity, the gene expression data should be analyzed using gene set analyses, or CDF plots that compare the entire catalog of expressed miR-221/222 predicted targets with all other genes as a group. The shift in expression of this group of genes might be as little as 5% or as much as 30% for a high abundance miRNA. This kind of analysis should be done for HSC, MPP1 and MPP2. This sentence about miR-221/222 targeets in the Discussion (lines 562-565) is badly off track. Fos was not shown to be a target in this study. Other targets were not shown to be unaffected. A proper analysis of miR-221/222 requirements for target gene repression could and should be provided, even if individual targets are not functionally linked to the observed emergent phenotypes of miR-221/222 deficiency.

8. The results in Figure 7 are striking. Are they reproducible in independent experiments? It appears that all replication shown in the figure are from recipients of the same transferred HSCs. An experiment of this importance bears repeating.

Rev. 2:

Jani et al present a novel mechanism for miRNA-221/222 in regulating the early progenitor compartment of the hematopoietic system- as well as providing important information about transcriptional alterations associated with isolation / purification stress on early progenitor cells. They evaluate the changes in number as well as the shifts in transcriptional profiles in stem and early progenitors after various stress and point to miRNA-221/222 as key players for the prevention of activation of these early primitive cells. They utilize a model where miRNA-221/222 is conditionally ablated in Vav-expressing cells (hematopoietic cells). While not critical to the findings, as the crosses still will ablate the miRNA 221/222 expression, it is a bit concerning that the authors have described this as a locus on the Y chromosome, but it appears to be located on the X chromosome. There are interesting and relevant data regarding stress and activation of early progenitors cells that would be relevant the stem cell community.

Major issues

1) The authors have presented this as a sex specific phenomenon (with several references to the miRNA 221/222 locus being on the Y chromosome. The crosses do work (regardless of if the locus is on the X or Y chromosome), but do clarify the location of the locus. As this is located on a sex chromosome the authors should examine if female mice progenitor cells respond differently to the stress (relevant with either X or Y location)

2) The stress response of the HSCs reported, looks similar to the "activated" state of stem cells reported by Cabezas-Wallscheid (doi.org/10.1016/j.cell.2017.04.018) and the authors should compare the scRNA profiles generated with this data set-

3) The animals used in these experiments were quite young for evaluating adult HSCs and in some cases there appear to be "batch effects from the mice- could the authors clarify age / sex had any role in the variation seen?

4) The authors present that the miRNA 221/222 depleted cells fail to reconstitute recipient in serial transplantations, however these HSCs are able to long term reconstitute PB, at much lower levels, but also reconstitute the early hematopoietic progenitor compartment robustly. This, together with the myeloid biased output, are similar to phenotypes of aged stem cells- and thus the loss of miRNA 221/222 could lead to increased activation of the stem cell compartment, leading to age-associated phenotypes. The authors should explore this alternative - some suggestions would be to compare profiles of the HSCs after transplant with aged HSCs (understandable if these cells are not available) look for altered CD150 expression levels in the miRNA 221/222 deficient vs proficient etc…

Minor Issues:

1) The introduction and discussion could benefit from some editing to make them more streamlined to ensure the main points don't get diluted.

2) Please clarify that cell surface markers used throughout for the isolation of cells- in some places it appear CD34- Flk2- was also included while the text indicates LSK CD150+ CD48- was used.

Rev. 3:

In their manuscript, Dr. Peter K. Jani et al. describe the role of miR-221/222 in HSCs and MPPs, highlighting novel potential targets. Overall, the results are interesting, important and novel, if confirmed. However, there are some issues that need to be addressed, as detailed below:

1) How long does the perturbation of the steady-state hematopoiesis due to "in vivo" social stress last? Is it reversible? This is important to evaluate the relevance of the observation.

2) Fig 1A, please specify if the figure refers to WT (miR-221/222 proficient) cells. In the same paragraph, Supplementary Fig.1 is quoted (line 151), but it is a bit confusing to see at this point results on Proficient/Deficient cells, while trying to evaluate what is written in the text referred to proficient cells only. Maybe separate figures would be easier to read. Moreover, I strongly suggest to highlight only significant p-values.

3) Line 150: according to Supplementary Fig.1, also MPP4 increase upon social stress. Please correct the text.

4) Line 160: the authors write that Paired t-test was used to evaluate statistical significance. It is not clear to me how the pairing was made, since the comparison was made between stressed and not stressed mice, not among the same mouse before and after stress. Please explain. In addition, was a normality test applied before choosing ttest? Please specify.

5) Supplementary Table 1 is quoted in different parts of the manuscript; however, it is not clear which tab one has to evaluate. An explanation of the Table and of the different tabs is necessary in my opinion. Importantly, I don't see any reference to where the raw data have been deposited in a public repository.

6) I have some difficulties in understanding the approach of using ActD prior to RNAseq, since there is already an internal control of the experiment, for example the equally "ex vivo/in vitro" treated miR-221/222 proficient cells. However, I realize that it allowed to highlight genes that would have not been picked up with other approaches. It would be helpful if the explanation of the reason for choosing ActD-treated cells was more convincing.

7) Supplementary Fig 4A: statistics is missing.

8) Line 285: MPP3 are also increased.

9) Lines 340-341: this assertion is too strong.

10) The lack of a functional validation of at least some of the putative mir-221/222 cluster indirect targets (in other words, of the transcriptional changes that occur in the absence of mir-221/222, seen here with RNA sequencing) is a strong limitation.

11) Lines 557-559: "we have identified…". The fact that miR221 might play an important role in HSPCs had been suggested in at least two previous papers (Felli N et al. PNAS 2005; Crisafulli L et al., Haematologica 2019). In particular, the differences observed here compared to previous publications should be discussed, with some hypothesis to explain discrepancies. For example, in Felli et al., which the authors quote, the demonstrated miR-221/222 direct target cKit does not appear as differentially expressed in the present manuscript. In Crisafulli et al., the mature form of miR-221 was found to be only minimally expressed in HSCs, and strongly upregulated in MPP1.

12) Too many assumptions in the Discussion (which is a bit too long), considering the absence of a validation for all the quoted genes.

13) The resolution of most figures is not good. In particular, in Fig. 1B gene names are almost impossible to read.

14) The manuscript is not particularly fluent and a bit difficult to read; there are some grammar mistakes. In addition, it is a bit confusing the fact that results obtained with miR-221/222 deficient cells appear later in the manuscript. I would rather describe deficient mice first, and then compare the results with those obtained with stressed WT mice, thus changing the structure of the manuscript. This sequence is the one that the authors followed in the Abstract. Moreover: the introduction is too long; the first paragraph of the Results is redundant, everything has been written in the Intro.

Minor issues:

- There are three acronyms in the title.

- Lines 255-267: it is a copy and paste of lines 243-254.

---

## [Decision Letter · Decision Letter 2]

17 Aug 2023

Dear Dr Jani,

Thank you for your patience while we considered your revised manuscript entitled "Stress or miR-221/222-deficiency both activate hematopoietic stem cells either to cell cycle entry or to upregulation of immediate early gene expression and miR-deficiency combined with stress enhances granulopoiesis, leading to loss of multipotency" for publication as a Research Article at PLOS Biology. This revised version of your manuscript has been evaluated by the PLOS Biology editors, the Academic Editor and two of the original reviewers.

Based on the reviews, we are likely to accept this manuscript for publication, provided you satisfactorily address the data and other policy-related requests stated below.

In addition, titles have to be succinct, direct and widely accessible, thus we would like you to consider a suggestion to improve it:

"The miR-221/222 cluster regulates hematopoietic stem cell quiescence and multipotency by suppressing both Fos/AP-1/IEG pathway activation and stress-like differentiation to granulocytes"

As you address these items, please take this last chance to review your reference list to ensure that it is complete and correct. If you have cited papers that have been retracted, please include the rationale for doing so in the manuscript text, or remove these references and replace them with relevant current references. Any changes to the reference list should be mentioned in the cover letter that your revised manuscript.

We expect to receive your revised manuscript within two weeks. 

*Published Peer Review History*

*Press*

Sincerely,

Ines

--

Ines Alvarez-Garcia, PhD

Senior Editor

PLOS Biology

DATA POLICY:

Many thanks for sending the data file containing some of the data underlying the graphs shown in the figure. I have checked them and I am missing data. Here is the full list of the data we need:

Fig. 1A-D; Fig. 2A-D; Fig. 3A-F; Fig. 4B; Fig. 5A-D, F; Fig. 6A-C; Fig. 7A-D; Fig. S1A-C; Fig. S2A-B; Fig. S3A-B; Fig. S4A-B; Fig. S5A-E and Fig. S6A-D

These include figures containing FACS data, but apart from a picture showing the successive plots and gates that were applied to the FCS files to generate the figure, we ask that you provide FCS files. If these are too big, you can deposit them for free in the Flow Repository (http://flowrepository.org/).”

Please also ensure that figure legends in your manuscript include information on where the underlying data can be found.

In addition, please make the data you have deposited in the NCBI database publicly available.

BLURB

Please also provide a blurb which (if accepted) will be included in our weekly and monthly Electronic Table of Contents, sent out to readers of PLOS Biology, and may be used to promote your article in social media. The blurb should be about 30-40 words long and is subject to editorial changes. It should, without exaggeration, entice people to read your manuscript. It should not be redundant with the title and should not contain acronyms or abbreviations. For examples, view our author guidelines: https://journals.plos.org/plosbiology/s/revising-your-manuscript#loc-blurb

Reviewers’ comments

Rev. 1: Mark Ansel - please note that this reviewer has signed his review

The revised manuscript adequately addresses each of my prior concerns. The additional data provided in the response to reviewers document, regarding predicted miR-221/222 target gene expression in wildtype and miR-221/222-deficient HSC and MPP cells, would be useful for all readers. I recommend including it in the manuscript. From the violin plots, it does appear that the predicted targets are slightly shifted in their expression in miR-221/222 deficient cells as predicted. CDF plots would probably make this clearer, but in any case these results help to clarify the direct versus indirect effects of miR-221/222. With regard to Fos, I agree with the authors handling of the discussion of this gene, which may indeed be both a direct AND indirect target in this case. Finally, I am sympathetic to the difficulty of repeating the serial transplantation experiments. Nevertheless, I do recommend that readers treat those results with caution, since they have not been replicated in an independent experiment.

Rev. 3:

I thank Dr. Peter K. and colleagues for addressing all the issues that I have raised, as well as the ones raised by the other reviewers. The manuscript has substantially improved.

---

## [Editor Report · Decision Letter 3]

16 Oct 2023

Dear Dr Jani,

Thank you for the submission of your revised Research Article entitled "The miR-221/222 cluster regulates hematopoietic stem cell quiescence and multipotency by suppressing both Fos/AP-1/IEG pathway activation and stress-like differentiation to granulocytes" for publication in PLOS Biology. On behalf of my colleagues and the Academic Editor, Connie Eaves, I am delighted to let you know that we can in principle accept your manuscript for publication, provided you address any remaining formatting and reporting issues. These will be detailed in an email you should receive within 2-3 business days from our colleagues in the journal operations team; no action is required from you until then. Please note that we will not be able to formally accept your manuscript and schedule it for publication until you have completed any requested changes.

PRESS

Sincerely, 

Ines

--

Ines Alvarez-Garcia, PhD

Senior Editor

PLOS Biology
